# Autophagy regulates the maturation of hematopoietic precursors in the embryo

Yumin Liu [1,4], Linjuan Shi [1,4], Yifan Chen [1,4], Sifan Luo [1,4], Yuehang Chen [1,4], Hongtian Chen[1], Wenlang Lan[1], Xun Lu [1], Zhan Cao[1], Zehua Ye [1], Jinping Li [1], Bo Yu[2], Elaine Dzierzak [3] & Zhuan Li [1] ✉

An understanding of the mechanisms regulating embryonic hematopoietic stem cell (HSC) development would facilitate their regeneration. The aorta-gonad-mesonephros region is the site for HSC production from hemogenic endothelial cells (HEC). While several distinct regulators are involved in this process, it is not yet known whether macroautophagy (autophagy) plays a role in hematopoiesis in the pre-liver stage. Here, we show that different states of autophagy exist in hematopoietic precursors and correlate with hematopoietic potential based on the LC3-RFP-EGFP mouse model. Deficiency of autophagy-related gene 5 (Atg5) specifically in endothelial cells disrupts endothelial to hematopoietic transition (EHT), by blocking the autophagic process. Using combined approaches, including single-cell RNA-sequencing (scRNA-seq), we have confirmed that Atg5 deletion interrupts developmental temporal order of EHT to further affect the pre-HSC I maturation, and that autophagy influences hemogenic potential of HEC and the formation of pre-HSC I likely via the nucleolin pathway. These findings demonstrate a role for autophagy in the formation/maturation of hematopoietic precursors.

Hematopoietic stem cells (HSC) are at the apex of the adult hematopoietic hierarchy, providing all mature hematopoietic cells and multipotent progenitor cells. In the embryo, the functional HSCs are generated in the aorta-gonad-mesonephros (AGM) region[1,2] and other sites[3–5]. Functional HSCs derived from hemogenic endothelial cells (HECs)[6,7], undergo developmental processes through (pre)HEC, pre-HSC I, and pre-HSC II stages[8–12]. Subsequently, HSCs colonize in the fetal liver and migrate into bone marrow[1].

Accumulating data have shown that the production of HSC is regulated by the different transcription factors[12–15], signaling pathways[16–18], closely interacting cells such as macrophages[19,20], and other regulators[1,21–23]. The continuum of developmental processes from HEC to hematopoietic precursors is described by transcriptome analysis. Bioinformatics approaches have aided in the identification of more surface markers to enrich these cells during the endothelial to hematopoietic transition (EHT) process. For example, CD44[9], Procr[24], CD27[25], Ace[8], combined with other basic markers (CD31, CD41, CD45) are able to enrich HECs and hematopoietic precursors. However, our knowledge about the meticulous mechanisms involved in HSC emergence and maturation remains incomplete.

Autophagy is one of the highly conserved intracellular degradation processes involved in maintaining cell survival, cell differentiation, cell death and is associated with various diseases[26]. Distinct types of autophagy have been identified, including chaperone-mediated autophagy, microautophagy and macroautophagy, the latter is commonly called autophagy (in this study, we hereafter refer to it as autophagy)[27]. Autophagy occurs in all cell types responding to environmental stresses, such as deprivation, oxidative stress and thereby maintains cell survival. Autophagy is initiated with the phagophore formation, where cytoplasm and cytoplasmic organelles are

[1]Key Laboratory of Functional Proteomics of Guangdong Province, Department of Developmental Biology, School of Basic Medical Sciences, Southern Medical University, Guangzhou, China. [2]Institute of Hematology, School of Medicine, Jinan University, Guangzhou, China. [3]Centre for Inflammation Research, The University of Edinburgh, Edinburgh, UK. [4]These authors contributed equally: Yumin Liu, Linjuan Shi, Yifan Chen, Sifan Luo, Yuehang Chen. ✉e-mail: zhuanli2018@smu.edu.cn

enveloped into double-membrane bound as autophagosome, which then migrates to the lysosome forming the autolysosome for degradation[28]. Each step is tightly regulated by the autophagy-related proteins that are initially identified in yeast and have homologs in mammals. The autophagosome formation requires autophagy-related genes (Atg) to activate the evolutionarily conserved ubiquitin-like conjugation. During the fusion process of autophagosome to lysosome, the unconjugated form of Atg8/LC3 (LC3-I) is changed into the lipid form (LC3-II). The early autophagic vacuoles and autolysosomes are able to be distinguished by the elegant transgenic mouse model LC3-RFP-EGFP (LC3$^{R/G}$) based on the distinct pH sensitivities of red fluorescent protein (RFP, pK$_a$4.5) and enhanced green fluorescent protein (EGFP, pK$_a$5.9)[29].

Autophagy plays vital roles in the hematopoietic lineage output in adults[26]. Based on the knockout mouse model, Atg5 is involved in the pro-B maturation, T cell survival/proliferation and HSC function[30,31], and Atg7 is as an essential regulator of HSC maintenance and differentiation[32]. Furthermore, the focal adhesion kinase family interacting protein of 200 kDa (Fip200), as a component of ULK-Atg13-Fip200 complex, regulates HSC function, and erythroid/myeloid differentiation in the fetal liver[33]. Recently, transcriptomic data have shown Atg5/7 is highly expressed in the pre-HSCs[34]. However, it is not known whether autophagy regulates HSC development in the earlier embryonic, pre-liver stage, particularly in the first HSC emergence in the AGM region.

Nucleolin (Ncl) is a multifunctional phosphoprotein that is ubiquitously distributed in various eukaryotic cell compartments, such as nucleolus, cytoplasm, and cell membrane. Ncl is directly involved in the regulation of gene transcription, RNA metabolism and DNA damage repair[35]. Ncl promotes the activity of transcriptional regulators HOXA9 and ERG to maintain stemness of human HSC[36]. Additionally, Ncl keeps G0 HSC in the mouse bone marrow[37]. Meanwhile, Ncl induces autophagy-dependent cell death/apoptosis through AMPK-autophagy axis in tumor cells[38].

In this study, by using Atg5 conditional knockout and autophagy reporter mouse model LC3$^{R/G}$, we examined the regulatory roles of autophagy in hematopoietic development. Our results reveal that Atg5 is essential for the formation/maturation of hematopoietic precursors and HSC function. Based on combined approaches, including single-cell RNA-sequencing data (10x genomics) and rescue experiments, we show that Atg5 deletion interrupts the temporal order of hematopoietic development and affects the maturation of pre-HSCs via Ncl pathways. These data suggest a role for autophagy in the regulation of HSC development in the embryonic AGM region.

## Results

### Autophagy is involved in hematopoietic development in the embryo

The LC3$^{R/G}$ mouse model is used to investigate the dynamics of autophagy in adults[29]. We have checked the expression of GFP and RFP in the HSCs of adult bone marrow (Supplementary Fig. 1a), which is similar to the recent report[39]. In the embryo, autophagic status (RFP$^+$GFP$^+$ represents autophagosome before fusion with autolysosome and RFP$^+$GFP$^-$ represents autolysosome and RFP$^-$GFP$^-$ represents no autophagic activity) was distinguished based on the fluorescence level of RFP and GFP (Fig. 1a–e). We found that the pattern of RFP and GFP expression appeared a bit different in the fetal liver from bone marrow (Fig. 1b, c and Supplementary Fig. 1a, b), the RFP signals were weaker compared to GFP signals. In the E12.5 LC3$^{R/G}$ fetal liver, more than 85% HSCs (Lin$^-$Sca1$^+$Mac1$^{low}$CD201$^+$, HSC) and hematopoietic stem/progenitor cells (HS/PC, Lin$^-$Sca1$^+$Mac1$^{low}$, LSM) were RFP$^+$GFP$^+$, much higher than that in the Lin$^-$ cells, similar to RFP$^+$GFP$^-$ cells, and verified by the opposite trend of RFP$^-$GFP$^-$ cells (Fig. 1c). Meanwhile, similar trends were found by only analysis of GFP fluorescence signals (Supplementary Fig. 1c, d), implying the existence of distinct

autophagic statuses is related to stemness/differentiation of HS/PC in the fetal liver.

Hematopoietic clusters emerged from HECs and include pre-HSC I (I Pre, CD31$^+$CD41$^{low}$CD45$^-$) and pre-HSC II (II Pre, CD31$^+$CD45$^+$). In the E11.5 AGM region, ~82% of pre-HSC I were RFP$^+$GFP$^+$, which was significantly higher than the percentage in endothelial cells (CD31$^+$CD45$^-$CD41$^-$, EC) and pre-HSC II (51.0% ± 2.2% and 37.4% ± 3.5%, respectively), and then the trend was in contrast to the percentage of RFP$^+$GFP$^-$ cells, consistent to the detection of GFP$^+$/GFP$^{low/-}$ cells. A quite low percentage of RFP$^-$GFP$^-$ cells were found in the pre-HSC I, much lower than that in the EC fraction and pre-HSC II (Fig. 1a, d, e and Supplementary Fig. 1e, f). The signals of GFP and RFP were also confirmed by the immunostaining in three fractions (RFP$^+$GFP$^+$, RFP$^+$GFP$^-$, and RFP$^-$GFP$^-$) of AGM ECs and pre-HSCs. Most RFP$^+$GFP$^+$ cells are positive for both GFP and RFP (Fig. 1f and Supplementary Fig. 1g, h), similar to the purity of flow cytometric sorting.

To study the specificity of LC3$^{R/G}$ for labeling autophagy status, immunostaining between GFP/RFP and p62/Lamp1 was performed. GFP, RFP, and p62 were colocalization in the EC and pre-HSC I and pre-HSC II (presenting autophagosome), and Lamp1 was mainly co-localized with RFP, but not with GFP (presenting autolysosomes) (Fig. 1f–g and Supplementary Fig. 1g, i, j), consistent with the previous report[40]. The geometric mean fluorescence intensity (GeoMFI) of GFP was highest in pre-HSC I compared to that in EC and pre-HSC II fractions, whilst RFP GeoMFI was lowest in the EC fractions of LC3$^{R/G}$ AGM region. Expectedly, the highest GeoMFI ratios of GRP/RFP were in the pre-HSC I, in line with the trend of RFP$^+$GFP$^+$ cells and GFP$^+$ cell percentage (Fig. 1e and Supplementary Fig. 1f and k), indicating the alteration of autophagy status.

To further investigate the potential of hematopoietic-related cells in different autophagic statuses, RFP$^+$GFP$^+$ and RFP$^{+/-}$GFP$^-$ cell fractions were cultured in the methylcellulose (because few RFP$^-$GFP$^-$ cells were obtained, cultures were performed by combined RFP$^+$GFP$^-$and RFP$^-$GFP$^-$ cells). Colony-forming unit cultures (CFU-C, including CFU-GEMM, CFU-GM, BFU-E, and CFU-E) were enriched in the RFP$^+$GFP$^+$ fractions, with a higher number of total CFU-Cs and CFU-GM, compared with RFP$^{+/-}$GFP$^-$ group (Supplementary Fig. 1l). Then, RFP$^+$GFP$^+$ and RFP$^{+/-}$GFP$^-$ cells were cocultured with OP9-DL1. The number of CD45$^+$ cells derived from RFP$^+$GFP$^+$ pre-HSC I and pre-HSC II was much higher than that from RFP$^{+/-}$GFP$^-$ groups, but not in the EC fractions (Fig. 1h), suggesting that the maturation of hematopoietic precursors (pre-HSC I and pre-HSC II) are relevant to the earlier autophagic process.

Explant culture is useful for studying hematopoietic precursor development. To check whether autophagy affects hematopoiesis, 3-methyladenine (3-MA, one of earlier stage autophagy inhibitors)[26] was added in the AGM explant (AGM$^{ex}$) cultures. Inhibition of autophagy resulted in the significant decline of total colony-forming unit culture (CFU-C) number (including CFU-GEMM, CFU-GM, BFU-E, and CFU-E) from E10.5-E11.5 AGM$^{ex}$, with the decrease of CFU-GM and BFU-E (Supplementary Fig. 1m). Furthermore, since 3-MA treatment failed to change the viable cell number, similar to the other autophagic inhibitor (chloroquine (CQ) and bafilomycin A1 (Baf1)) (Supplementary Fig. 1n), 3-MA reduced the percentage of pre-HSC II, but not of EC and pre-HSC I in E11.5 AGM$^{ex}$. Meanwhile, the absolute numbers of EC, pre-HSC I and II were reduced (Supplementary Fig. 1o, p). Then, the inhibitors of late autophagy (CQ and Baf1) block autophagosome-lysosome fusion in the explant cultures. For 12-hour explant cultures, the existence of chloroquine (CQ, 5 μM) decreased the percentages and cell numbers of pre-HSC I and pre-HSC II, but not in the endothelial cells (Supplementary Fig. 1q, r), and so did Baf1 (100 nM, Supplementary Fig. 1s, t). Even if the treatment of Baf1 existed for 6 hours, the reduction of pre-HSC I in percentage and cell number was obvious (Supplementary Fig. 1u, v). So, both CQ and Baf1 had similar effects to 3-MA on AGM explant cultures. These ex vivo data demonstrate the involvement of autophagy in hematopoietic precursor development.

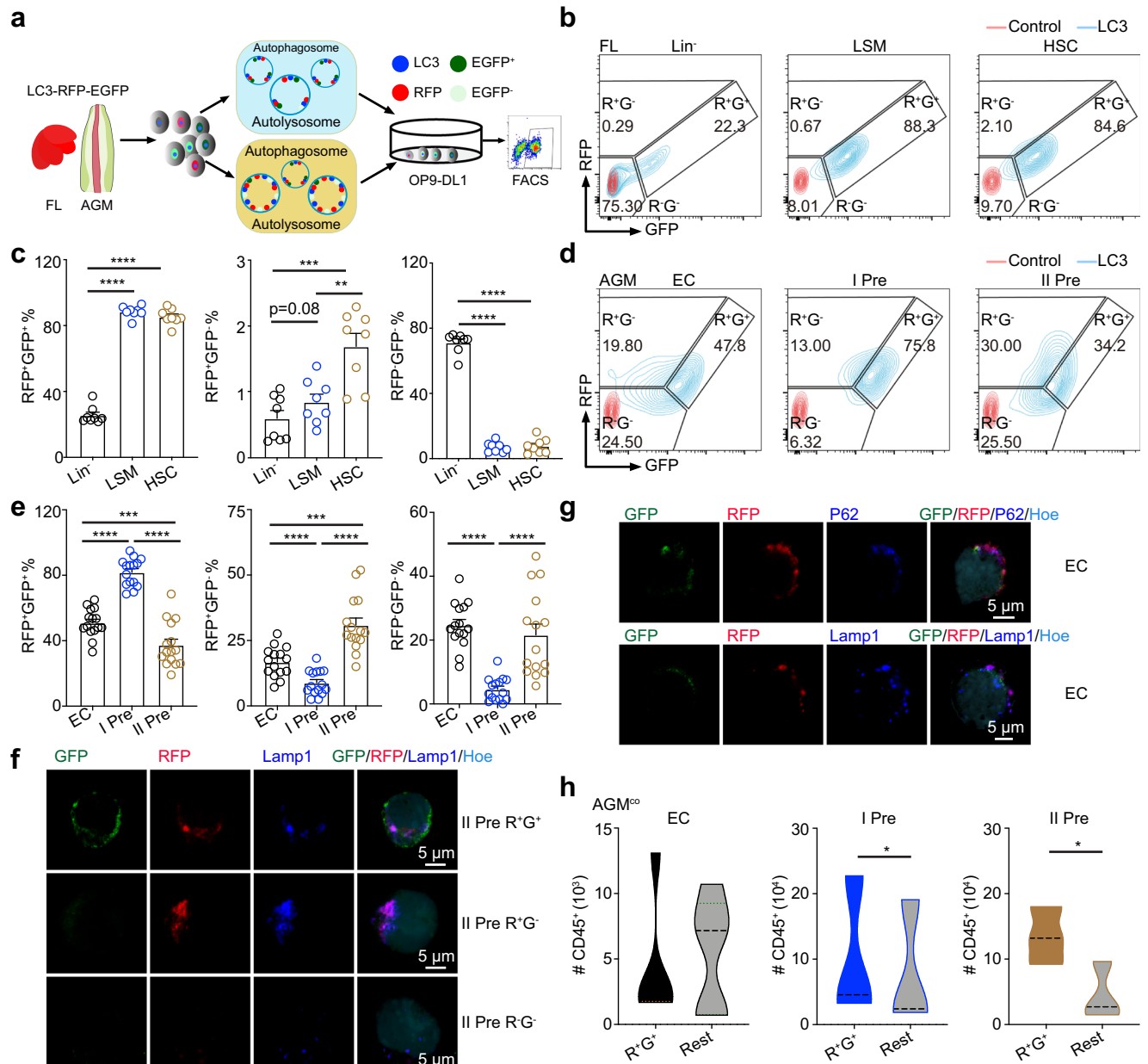

**Fig. 1 | Identifying the autophagic status in E11.5 LC3-RFP-EGFP (LC3[R/G]) AGM and E12.5 Fetal liver (FL) cells. a** The schematic for detecting autophagic status. Adobe Illustrator software was used for drawing this figure. **b** Flow cytometric analysis showing the RFP and GFP fluorescence level in Lin⁻, Lin⁻Sca1⁺Mac1[low] (LSM) and CD201⁺LSM (HSC) cells of E12.5 LC3[R/G] fetal liver. Red line = WT control, blue line = LC3-RFP-EGFP (LC3[R/G]). **c** The percentage of RFP⁺GFP⁺, RFP⁺GFP⁻ and RFP⁻GFP⁻ in distinct cell fractions (Lin⁻, LSM and HSC) of fetal liver. Error bars represent mean ± SEM. *n* = 8 biologically independent embryos, **\**p = 0.0012, ***\**p = 0.0002, ****\**p < 0.0001. **d** Presentative flow data displaying the RFP and GFP fluorescence level in control viable cells and endothelial cells (EC, CD31⁺CD41⁻CD45⁻), pre-HSC I (I Pre, CD31⁺CD41[low]CD45⁻) and pre-HSC II (II Pre, CD31⁺CD45⁺) of E11.5 LC3[R/G] AGM region. WT viable cells as a negative control. Red line = Control, blue line = LC3[R/G]. **e** The percentage of RFP⁺GFP⁺, RFP⁺GFP⁻ and RFP⁻GFP⁻ cells in distinct cell fractions of LC3[R/G] AGM region. Error bars represent

mean ± SEM. *n* = 15 biologically independent embryos, ***\**p < 0.001, ****\**p < 0.0001. **f** Representative immunostaining data showing the signals of GFP, RFP, and Lamp1 in the RFP⁺GFP⁺, RFP⁺GFP⁻ and RFP⁻GFP⁻ pre-HSC II cells. GFP = Green, RFP = Red, Lamp1 = Blue and Hoechst = light blue. Scale bar = 5 μm. Cells from 3 independent experiments. **g** Representative immunostaining data showing the colocalization of GFP, RFP, and p62/Lamp1 in the EC. GFP = Green, RFP = Red, P62 = Blue, Lamp1 = Blue, and Hoechst = light blue. Scale bar = 5 μm. Cells from 3 independent experiments. **h** The production of CD45⁺ cells derived from both RFP⁺GFP⁺ and the rest subfractions (including RFP⁺GFP⁻ and RFP⁻GFP⁻) in EC, I Pre, II Pre populations of LC3[R/G] AGM cocultured with OP9-DL1 (AGM[co]). *n* = 5, 3, and 3 biologically independent experiments in EC, I Pre, and II Pre, respectively, *\**p < 0.05. Rest = RFP⁺GFP⁻ and RFP⁻GFP⁻ cells. Statistical significance was determined by one side Student's t-test.

## Atg5 regulates HS/PC function in the embryo

Autophagy-related genes regulate HSC maintenance and differentiation[31,33]. Since Atg5 is the key regulator of autophagy, to test the function of Atg5 in hematopoiesis in the embryo, we generated *Vec-Cre;Atg5[fl/fl]* (KO) and control embryos (*Vec-Cre⁻;Atg5[fl/fl]* or *Atg5[fl/+]*, Ctr). QRT-PCR was performed to confirm the deletion efficiency of

*Atg5* in the ECs (Supplementary Fig. 2a). Methylcellulose cultures revealed that Atg5 deletion resulted in a 38% reduction in CFU-C number per E12.5 fetal liver, with the decrease mainly in CFU-E. Also, CFU-C frequency from the same input cells (1000 cells) was reduced significantly, indicating autophagy is involved in mediating the development of HPC (Fig. 2a, b), in line with a previous report[33]. Flow

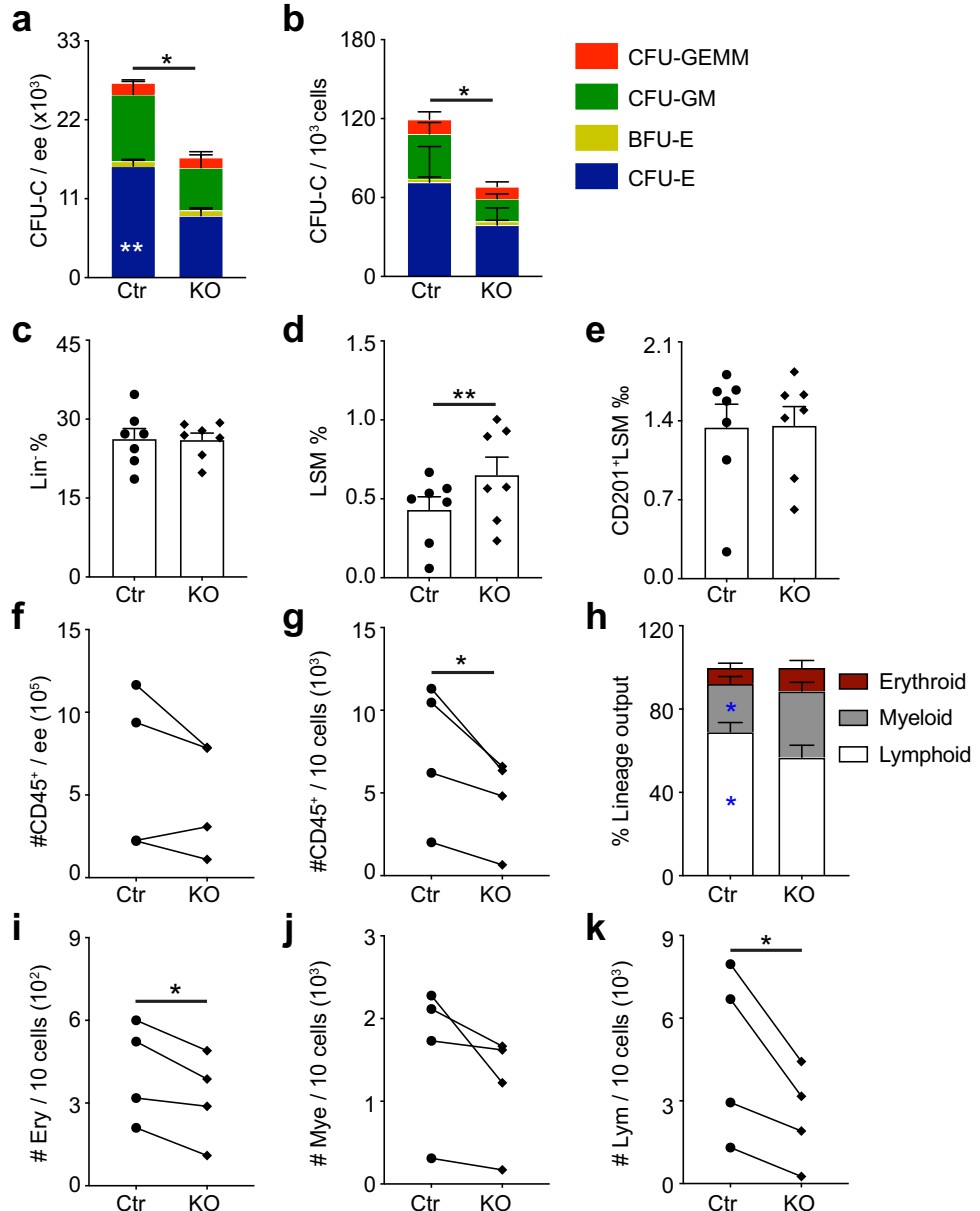

**Fig. 2 | Atg5 deletion changes hematopoietic progenitor cell function and HSC differentiation in the fetal liver. a**, **b** Methylcellulose culture data showing the number of CFU-Cs and number of each hematopoietic colony type in the E12.5 fetal liver (indicated by color bars) per embryo equivalent (ee) or per 1000 input cells of control and KO. Error bars represent mean ± SEM. $n = 4$ biologically independent embryos, *$p < 0.05$, **$p = 0.0016$. **c**–**e** Flow cytometric analysis showing the percentage of Lin⁻, Lin⁻Sca1⁺Mac1^low (LSM) and CD201⁺LSM (HSC) in E12.5 fetal liver cells. $n = 7$ biologically independent embryos, **$p = 0.006$. Circle = control (Ctr), inverted square = KO. **f** The number of CD45⁺ hematopoietic cells from E12.5 fetal

liver (FL) HSCs was unchanged. $n = 4$ biologically independent experiments. **g** Atg5 deletion resulted in the reduced hematopoietic differentiation capacity of HSC in the E12.5 FL HSC. $n = 4$ biologically independent experiments. *$p = 0.02$. **h** The lineage output of HSC in E12.5 KO FL compared with control group. Error bars represent mean ± SEM. $n = 4$ biologically independent experiments. *$p < 0.05$. **i**–**k** The number of erythroid, myeloid and lymphoid cells from the same input HSCs (10 cells) in E12.5 KO FL compared with control group. $n = 4$ biologically independent experiments, *$p < 0.05$. Circles = control (Ctr), inverted squares = KO. Statistical significance was determined by one side Student's t-test.

cytometric analysis of E12.5 fetal liver cell subsets showed the percentage of LSM was increased, whereas the frequency of Lin⁻ cell and HSC was unchanged (Fig. 2c–e). Meanwhile, OP9 cocultures showed that hematopoietic cell (CD45⁺ cell) number derived from KO FL HSCs was unchanged compared with control group. However, the obvious reduction of hematopoietic cells was found from the same input HSCs, with a decrease of erythroid and lymphoid differentiation ability and altered lineage output (Fig. 2f–k), implicating a possible role for autophagy in hematopoiesis.

Therefore, to investigate the role of autophagy in HSC emergence, distinct stages of AGM regions were analyzed. The percentages and

numbers of CD41⁺CD45⁻ cells failed to change from E9.5–E11.5 KO AGM region compared to the control. Furthermore, the frequency and the number of CD41⁻CD45⁺ cells were reduced only in the E11.5 KO AGM region but not in the E9.5-E10.5 (Supplementary Fig. 3a–f). Meanwhile, the deficiency of Atg5 induced a significant decrease in CFU-C in the E10.5-E11.5 AGM region with the reduction of CFU-GM. This was a similar trend at E9.5 (Fig. 3a) and indicates that autophagy influences HPC development in the AGM region.

To check whether Atg5 influences HSC function, cells from KO and control E11.5 AGMs (CD45.2/2) were injected into irradiated recipients (CD45.1/1) and the chimerism of donor cells was examined at 4

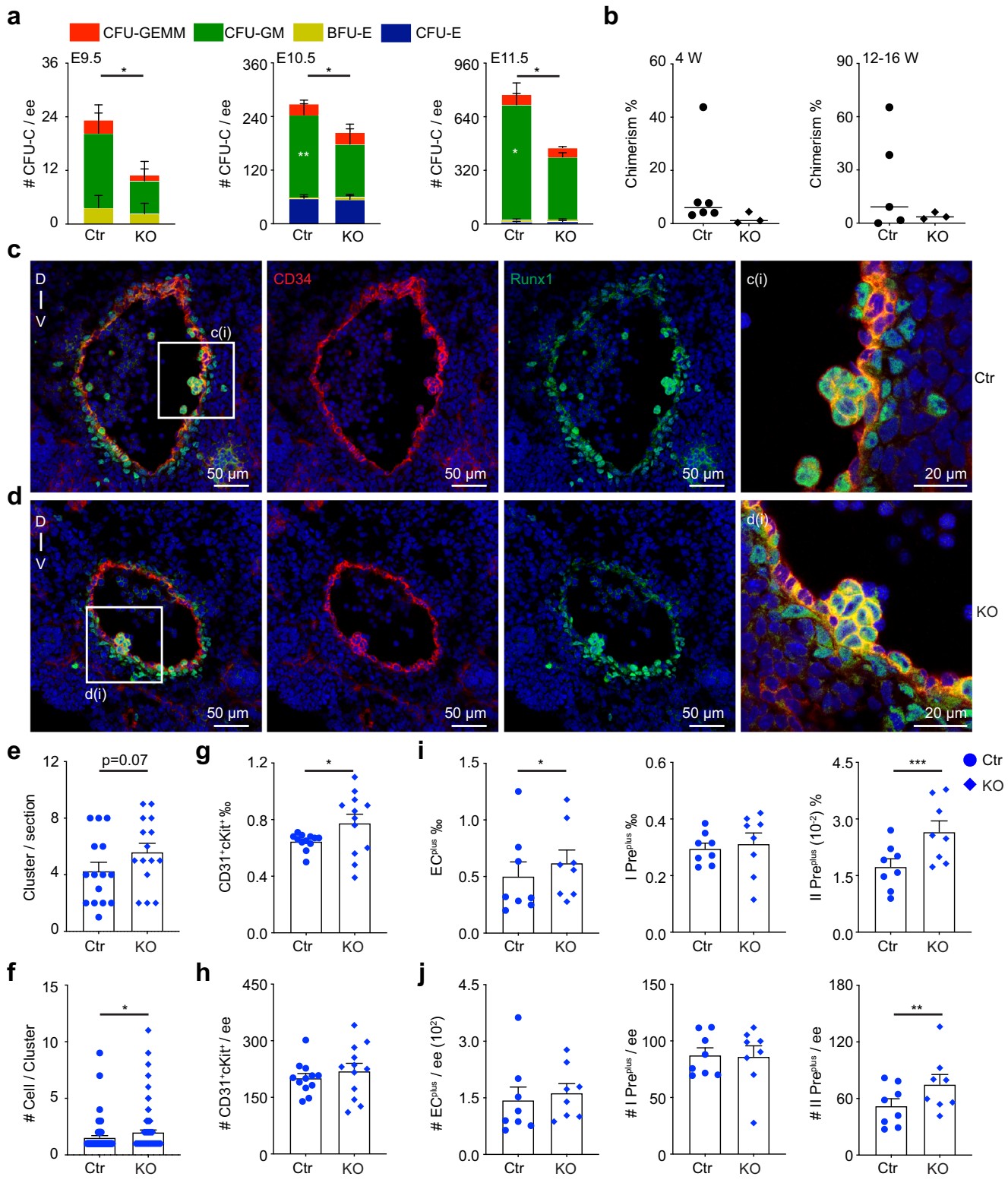

and 12-16 weeks post-injection. At 4 weeks, 3 out of 6 recipients were engrafted in the control group. In the Atg5 deficient AGM group, the chimerism of the 3 recipients was less than 5%. After 12-16 weeks of transplantation, the chimerism in the Atg5 deficient AGM recipients was 4.08 ± 1.14% compared with control transplant recipients with the chimerism of 22.92 ± 12.65% (Fig. 3b). These data show the effects of autophagy in the HS/PC activity in the AGM region.

**Atg5 deletion alters hematopoietic precursor formation/ maturation**

HS/PCs emerged from HECs in the AGM region, by forming hematopoietic clusters[1]. Immunostaining of CD34, Runx1 and cKit was performed to check hematopoietic clusters on cryosections from E10.5 control and KO embryos. CD34+Runx1+ hematopoietic clusters containing 1, 2, 3, 4, or ≥5 cells per cluster were quantitated. The number of

**Fig. 3 | Atg5 ablation resulted in the alteration of immature hematopoietic precursor cells and HSCs in the AGM region. a** The number of CFU-Cs and number of each hematopoietic colony type in the E9.5-E11.5 AGM (indicated by color bars). Error bars represent mean ± SEM. $n = 3$, 8 and 4 biologically independent experiments in E9.5, E10.5 and E11.5 AGM respectively, $*p < 0.05$, $**p = 0.0024$. **b** Percentage of donor cell (CD45.2/2) chimerism in peripheral blood of irradiated recipients receiving control or KO AGM region cells after transplantation 4 weeks and 12–16 weeks. Circles and inverted squares indicate individual recipients of control or KO cells, respectively. Each dot represents one recipient. $n = 3$ biologically independent experiments. **c, d** Representative immunostaining of cryosections in E10.5 control and KO AGM region. Red = CD34, Blue = Hoechst, Green = Runx1. **c** (i) and **d** (i) showing the hematopoietic cluster cells in the control and KO aorta. Scale bars are indicated. **e** The average number of CD34+Runx1+ hematopoietic clusters per section. Error bars represent mean ± SEM. $n = 15$ sections from 3 independent embryos, $p = 0.07$. Statistical significance was determined by Mann−Whitney U-test. **f** The average cell number in the detected CD34+Runx1+ hematopoietic clusters. Error bars represent mean ± SEM. $n = 15$ sections from 3 independent embryos, $*p = 0.0123$. Circle = control (Ctr), inverted square = KO. Statistical significance was determined by Mann−Whitney U-test. **g, h** The percentage (**g**) of hematopoietic clusters (CD31+cKit+) was increased and the cell number (**h**) was comparable in the KO AGM as compared to control. Error bars represent mean ± SEM. $n = 12$ biologically independent embryos, $*p = 0.015$. **i, j** The percentage (**i**) and cell number (**j**) of EC$^{plus}$ (CD31+CD41−CD45−CD44+CD201+cKit+DLL4+), pre-HSC I$^{plus}$ (I Pre$^{plus}$, CD31+ CD41$^{low}$CD45−CD201+cKit+) and pre-HSC II$^{plus}$ (II Pre$^{plus}$, CD31+CD45+CD201+cKit+) in the E11.5 AGM region. Error bars represent mean ± SEM. $n = 8$ biologically independent embryos, $*p = 0.0398$, $**p = 0.0036$, $***p = 0.0007$. Statistical significance was determined by one side Student's t-test unless the statistical test was indicated.

clusters containing 2 cells per section was significantly higher in the KO aorta than control, with the other cluster numbers per section trending to an increase (Fig. 3c–e and Supplementary Fig. 3g). The number of cells per cluster was enhanced in Atg5-deleted sections ($2.0 ± 0.20$ vs $1.53 ± 0.17$ cells/cluster) (Fig. 3f). The similar trends were found by staining cKit and CD34 (Supplementary Fig. 3h–j). In the E11.5 KO AGM region, the percentage of phenotypic hematopoietic cluster cells (CD31+cKit+) was increased, but the total number was comparable (Fig. 3g, h), possibly relative to the changed capacity of recruitment or migration of hematopoietic precursors, as is reported some cells of bigger hematopoietic clusters are recruited from circulation[41].

The regulation of autophagy on hematopoietic precursors was further examined in E10.5-E11.5 AGM cells. At the E10.5, the percentage of pre-HSC I was boosted approximately one-fold after Atg5 deletion ($1.20 ± 0.3‰$ vs $0.61 ± 0.1‰$), whilst the number was increased around 60% ($137 ± 23$ vs $86 ± 13$ cells/embryo equivalent, ee). EC and pre-HSC II cells were not changed. One day later, the alteration of pre-HSC I disappeared. However, the development of pre-HSC II was influenced, with a significant reduction in the percentage ($0.19 ± 0.016\%$ vs $0.24 ± 0.004\%$) and cell number ($793 ± 85$ vs $1049 ± 81$ cells/ee) (Supplementary Fig. 3k, l) and the hemogenic potential of ECs was reduced, whilst the hematopoietic ability of pre-HSC II was increased (Supplementary Fig. 3m, n), suggesting the dynamic effects of autophagy on the formation/maturation of hematopoietic cells.

More surface markers were added to further enrich EC$^{plus}$ (CD31+CD45−CD41−CD44+DLL4+CD201+cKit+), pre-HSC I$^{plus}$ (CD31+ CD45−CD41$^{low}$CD201+cKit+) and pre-HSC II$^{plus}$ (CD31+CD45+CD201+ cKit+). No alterations were found in the EC$^{plus}$ and pre-HSC I$^{plus}$, in line with that in the EC and pre-HSC I. The percentage and cell number of pre-HSC II$^{plus}$ was increased around 60% and 44% in the KO AGM region compared to the control, in contrast to the trend of pre-HSC II (Fig. 3i, j). Expectedly, the frequency and the number of pre-HSC II$^{minus}$ (pre-HSC II without pre-HSC II$^{plus}$, representing mature hematopoietic cells) was decreased more dramatically (Supplementary Fig. 3o), demonstrating that Atg5 deletion promotes pre-HSC emergence and blocks their maturation. Altogether, these data suggest that Atg5 regulates the formation/maturation of hematopoietic precursors into functional HS/PCs.

## Atg5 deficiency undermines autophagic process

To study whether inhibition of autophagy by Atg5 deficiency disrupts the autophagic process, we mated LC3$^{R/G}$-Vec-Cre;Atg5$^{fl/+}$ with Atg5$^{fl/fl}$ to gain LC3$^{R/G}$-Vec-Cre;Atg5$^{fl/fl}$ (LC3$^{R/G}$-KO) embryos. Firstly, the percentage of RFP+GFP+ cells was enhanced and that of RFP+GFP− cells was reduced in LC3$^{R/G}$-KO EC, pre-HSC I, pre-HSC II, and hematopoietic clusters (CD31+cKit+) compare to the corresponding fractions of LC3$^{R/G}$. The reduction of RFP−GFP− cells percentage was observed in the EC, pre-HSC II and hematopoietic clusters (Fig. 4a, b), indicating that Atg5 conditional deletion impairs the autophagy process of hematopoietic precursors. Secondly, Atg5 is reported to affect

autophagosome formation, the further effect of which is altering autophagosome-lysosome fusion. As the GFP fluorescence level is diminished during the process of autophagosome-lysosome fusion, we examined the GeoMFI of GFP and RFP. GeoMFI of GFP was increased significantly in the RFP+GFP+ fractions of EC, pre-HSC I, pre-HSC II, and hematopoietic clusters (CD31+cKit+) of KO AGM cells as compared with control (Fig. 4c, d). Although RFP GeoMFI of the total fraction was low, especially in LC3$^{R/G}$ ECs compared with other fractions, GFP GeoMFI was higher in the pre-HSC I as well as the ratios of GFP/RFP, comparable to the RFP+GFP+ or GFP+ percentage in the pre-HSC I (Fig. 1e, 4a and Supplementary Fig. 1f, 4a–h). However, no alteration of RFP GeoMFI was found in the RFP+GFP+ and RFP+GFP− fractions of EC, pre-HSC I and II of LC3$^{R/G}$-KO compared to LC3$^{R/G}$ (Fig. 4e). Additionally, the RFP+GFP+ percentage of EC were increased in 3-MA treatment AGM region (Supplementary Fig. 4i). The existence of Baf1 (blocking autophagosome-lysosome fusion) resulted in the increasing percentage of RFP+GFP+ cells of EC, pre-HSC I, pre-HSC II and hematopoietic clusters, even if only pre-HSC I cells were affected during 6-hour treatment (Supplementary Fig. 1u and 4j), with the suggestion that Atg5 affects the formation of autophagosome in hematopoietic related cells and may further impact the fusion of autophagosome and lysosome.

As it is reported that autophagy restricts mitochondrial activity[42], we measured mitochondrial mass by the MitoTracker Green (MTG) probe and mitochondrial activity by TMRE in control and Atg5 deficient AGM region. TMRE/MTG GeoMFI ratios were reduced significantly in the EC, pre-HSC II, but not in the pre-HSC I and hematopoietic cluster (CD31+cKit+) fractions (Fig. 4f and Supplementary Fig. 4k, l), showing that autophagic inhibition leads to the alteration of mitochondrial activity in the AGM region. Overall, Atg5 deficiency indeed blocks the autophagic process of hematopoietic-related cells to regulate HSC development.

## Profiling transcriptomic atlas between control and Atg5-deficient AGM cells

To illustrate the regulatory roles of autophagy in hematopoiesis, we used the droplet-based scRNA-seq (10X Chromium) method to accurately measure the gene expression profiles of individual cells (7AAD−Ter119− cells) in the E10.0 AGM (Table S1). 52836 cells passed rigorous quality control with no batch effect. An average of 4383 genes (500-7000) and 21046 transcripts (611-75049) were detected in each individual cell (Supplementary Fig. 5a).

Uniform manifold approximation and projection (UMAP) of all cells separated into 20 clusters by using Seurat software, and included EC (Ramp2, Cdh5), hematopoietic cells (HC, Ptprc, Tyrobp, Fcer1g), megakaryocytes (Mk, Pf4), mesenchymal cells (Mes, Pdgf, Cxcl12), neuron development-related (NPC, Neuron and Schwann cells, Sox2, Ascl1) and epithelial cell (EPC, Epcam) clusters. Hematopoietic-related cells (including EC and HC) were readily distinguished from other cells and the percentage of EC/HC was increased in the Atg5-deleted group

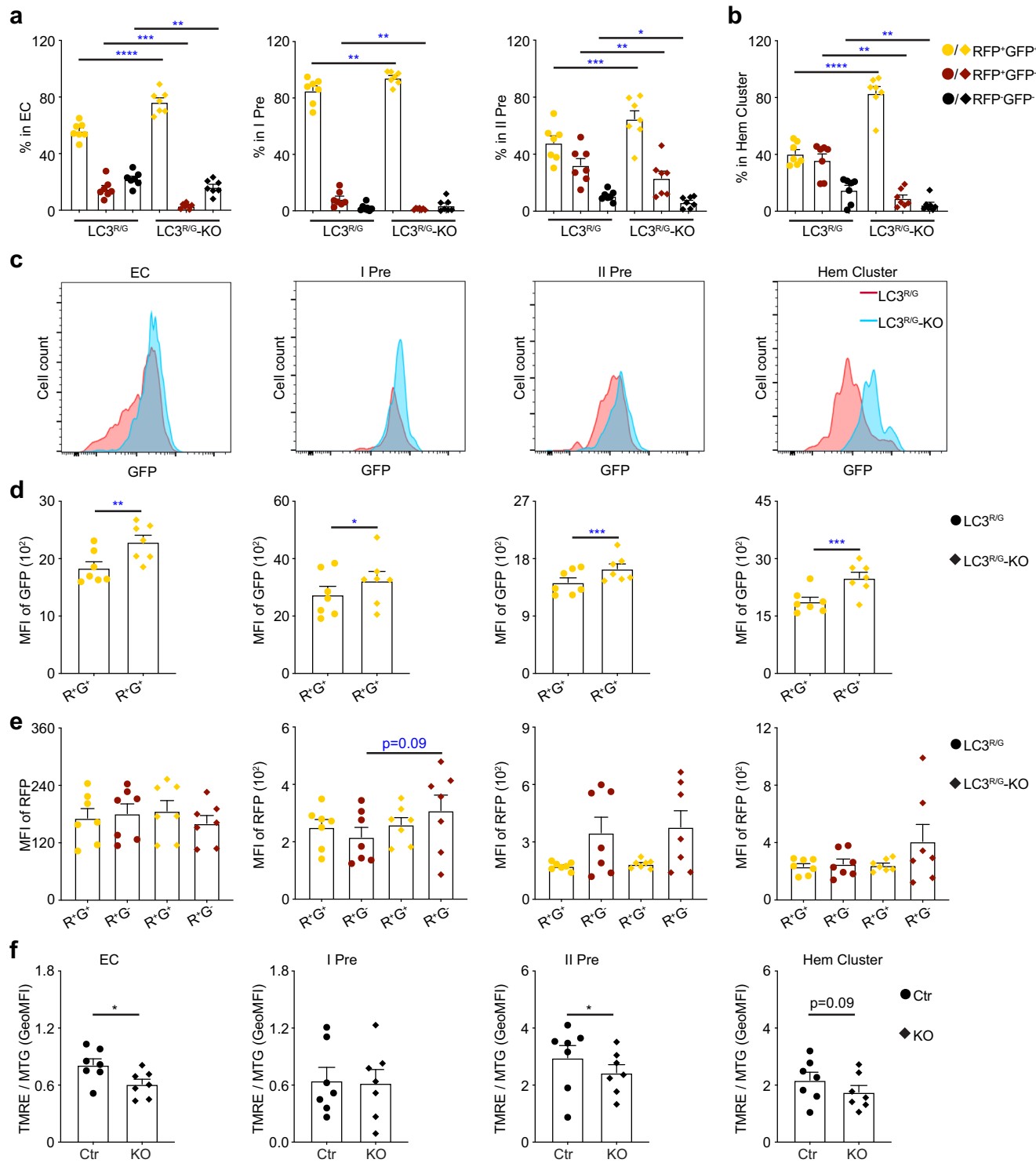

compared to the control group (Supplementary Fig. 5b–d). The cell cycle was changed in some KO clusters, such as hematopoietic cells, Mk and mesenchymal cells (Supplementary Fig. 5e), and is consistent with impaired hematopoietic development.

To gain insight into the regulatory mechanism of autophagy on hematopoietic development, the endothelial and hematopoietic cell transcriptomes were further clustered. Nine clusters, including venous EC (C1-C2, vEC: *Aplnr*, *Nrp2*), arterial EC (C3, aEC: *Gja5*, *Gja4*), HEC (C4, *Hlf*, *Runx1*, *Gfi1*), pre-HSC (C5, *Myb*, *Spn*), myeloid progenitor cells (C6, MPC) / macrophages (C7, Mac)(*Cybb*, *Tyrobp*), Mk (C8, *Gp1bb*, *Gp5*),

Erythroid (C9, *Gypa*, *Klf1*) were found to be separated. The Mk and erythroid clusters were distant from the other clusters. Cell proportions of vEC and aEC were changed in the KO group at the cost of Etv2+ EC (Fig. 5a, b and Supplementary Fig. 5f). Consistently, immunostaining for vascular vessel showed that the diameter and area of aorta were decreased in the KO group compared with control (Fig. 5c, d). Importatntly, Atg5 deficiency increased cell proportions of HEC and pre-HSC from selected EC and hematopoietic cell clusters, while the mature hematopoietic cell clusters (C6-C8) were reduced (Fig. 5a, b). Meanwhile, transcriptomic analysis displayed that the alteration of cell

**Fig. 4 | Inhibition of autophagy undermines autophagic process in hematopoietic-related cells of AGM cells. a** Atg5 deficiency led to an increased percentage of RFP⁺GFP⁺ cells and decreased the percentage of RFP⁺GFP⁻ cells in the EC, pre-HSC I and pre-HSC II fractions. Error bars represent mean ± SEM. $n = 7$ biologically independent embryos, *$p = 0.0155$, **$p < 0.01$, ***$p < 0.001$, ****$p < 0.0001$. Colored circle = LC3$^{R/G}$, colored inverted square = LC3$^{R/G}$-KO. Yellow represents RFP⁺GFP⁺ signal, dark red represents RFP⁺GFP⁻ signal and black presents RFP⁻GFP⁻ signal. **b** The frequency of RFP⁺GFP⁺, RFP⁺GFP⁻ and RFP⁻GFP⁻ cells in the hematopoietic clusters from E11.5 LC3$^{R/G}$-KO AGM region. Error bars represent mean ± SEM. $n = 7$ biologically independent embryos, **$p < 0.01$, ****$p < 0.0001$. Hem cluster = hematopoietic clusters (CD31⁺cKit⁺). Yellow represents RFP⁺GFP⁺ signal, dark red represents RFP⁺GFP⁻ signal and black presents RFP⁻GFP⁻ signal. **c** Representative histograms showing the GFP fluorescence level in the EC, pre-HSC I, pre-HSC II and hematopoietic clusters. Red line = LC3$^{R/G}$ group, blue line = LC3$^{R/G}$-KO group. **d** Comparison analysis of the geomean fluorescence intensity (GeoMFI) of GFP signals in the RFP⁺GFP⁺ EC, pre-HSC I, pre-HSC II and hematopoietic clusters of LC3$^{R/G}$ and LC3$^{R/G}$-KO. Error bars represent mean ± SEM. $n = 7$ biologically independent embryos, *$p = 0.0437$, **$p = 0.0013$, ***$p < 0.001$. **e** The GeoMFI of RFP in the different fractions of EC, pre-HSC I, pre-HSC II and hematopoietic clusters of the LC3$^{R/G}$ and LC3$^{R/G}$-KO. Error bars represent mean ± SEM. $n = 7$ biologically independent embryos. Circle = LC3$^{R/G}$, inverted square = LC3$^{R/G}$-KO. **f** The GeoMFI ratios of TMRE/MTG showing the reduction of mitochondrial activity in the EC, pre-HSC II, but not in the pre-HSC I and hematopoietic clusters of KO compared to control. Error bars represent mean ± SEM. $n = 7$ biologically independent embryos, *$p < 0.05$. Statistical significance was determined by one side Student's t-test.

cycle phase existed in mature hematopoietic cells and HEC/pre-HSCs, and flow analysis confirmed the active phase of cell cycle in the pre-HSC IIs, but not in EC or pre-HSC I (Fig. 5e, f and Supplementary Fig. 5g), implying the possible alteration of EHT and the maturation of hematopoietic cells.

### Transcriptomic alterations in the EHT process after Atg5 depletion

To clearly illustrate the transcriptional changes during EHT, the vEC, aEC, and pre-HSC profiles further separated into ten subclusters by known venous/arterial vascular endothelial, hemogenic, and hematopoietic genes, and showed vEC (C1-C4: vEC1-4), arterial EC (C5-C7: aEC1-3), HEC (C8), and pre-HSCs (C9-C10: pre-HSC I and II) (Fig. 5g, h). Increased endothelial cell proportions existed in the KO subclusters vEC4, aEC1 and aEC3, connecting arterial and venous endothelial cells from the UMAP visualization. Meanwhile, the proportion of HEC was comparable but the percentages of pre-HSC I and II were increased by Atg5 deficiency (Fig. 5g–i).

Trajectory analysis by Monocle 2 was performed at single-cell resolution to compare the temporal order of HEC/pre-HSC in the control and KO AGM regions. These data showed that Atg5 deletion led to the accumulation of pre-HSC I from EC to pre-HSC transition and a delay in the developmental process of pre-HSC I relative to the maturation of pre-HSCs, consistent with the flow analysis mentioned above (Fig. 3i, j, 6a, b, and Supplementary Fig. 3k, l). RNA velocity analysis estimated the spliced and unspliced gene state, for example, unspliced *Runx1* appeared higher in the KO HEC as well as Gfi1, indicating the block of EHT process. Unspliced *Kit* was higher in the KO pre-HSC I compared to control group, opposite to the trend of *Spn* (Fig. 6c and Supplementary Fig. 6a, b), suggesting that Atg5 probably promotes the process of pre-HSC development by regulating the spliced state of *Runx1*, *Gfi1* and *Kit*. To further check the cell components of HEC and pre-HSC during developmental trajectory, the developmental process of pre-HSC I in KO group was altered compared to the control (Fig. 6d and Supplementary Fig. 6c), in line with our flow analysis data. These data indicate the possible signaling pathways of autophagy in regulating EHT process.

### Atg5 depletion changes the hematopoietic-related biological process

Since we found the EHT process was affected upon Atg5 deletion, HEC and pre-HSC transcriptomic clusters were further analyzed for differentially expressed genes (DEGs) and gene ontology biological process (GOBP, Fig. 7a–i). Volcano plots showed up/down regulated genes in the KO HECs and the top 15 DEGs were displayed (Fig. 7a, Supplementary Data 1). The expression of enriched genes in KO HECs was linked to restricted Smad protein phosphorylation, in agreement with our previous report[17]. Control HECs showed genes enriched for response to IL-1, autophagosome maturation, positive regulation of ubiquitin transferase activity (which is the main pathway for autolysosome degradation[43]), and mitochondrial fission (Fig. 7g, Supplementary Data 2), related to the impairment of autolysosome formation and mitochondrial activity after Atg5 ablation.

Volcano plots and bar graphs showed up-regulated genes *Ptpn21*, *Atf5* and down-regulated gene *Sox9* in the KO pre-HSC I (Fig. 7b, e). GOBP analysis revealed gene enrichment in cell-cell adhesion and vasculogenesis processes in KO pre-HSC I, and enrichment in bone morphogenesis and antigen processing and presentation in control (Fig. 7h). Chemotaxis factors (*Ccl3*, *Cx3cr1*) were highly expressed in the control pre-HSC II, and mainly related to leukocyte migration. KO pre-HSC IIs were enriched in the process of nuclear division and angiogenesis (Fig. 7c, f and i, Supplementary Data 1–2). Furthermore, the KEGG signaling pathway analysis showed that pathways related with ubiquitin mediated proteolysis, oxidative phosphorylation were enriched in the control EC and pre-HSCs, whilst p53 signaling pathway and VEGF pathway were enriched in the KO pre-HSCs (Fig. 7j). These data, together with the results on the increased size of hematopoietic clusters, connect it with the impairment of migration in the KO group.

### Atg5 deletion influences EHT process through nucleolin pathway

Cellchat was used to compare the possible communication of all cells via ligand-receptor (L-R) expression. The number and strength of L-R expression/interaction was increased in the KO compared to control (Supplementary Fig. 7a, b). The Gas pathway was higher in the control, but the opposite trend of most other pathways existed in the KO group, including Grn (granulin), Mdk (midkine), Ptn (pleiotrophin) and Kit (Supplementary Fig. 7c), in line with SCF-dependent pre-HSC production[44]. Since Ncl is the receptor for Mdk and Ptn[45–47], the expression/interactions of Ptn/Mdk-Ncl were enhanced in KO HEC, pre-HSC I and II cells (Fig. 8a), indicating the possible involvement of Ncl in the hematopoiesis.

The expression of *Ptn* and *Mdk* was decreased in the EC of E10.5 KO AGM, but *Ncl* was highly expressed (Supplementary Fig. 7d). Furthermore, immunostaining data showed Ncl with a sheet morphology in the cytoplasm, localized between the nucleus and cell membranes in control EC and pre-HSCs. Ncl fluorescence signals were reduced in KO EC, but not in pre-HSCs and confirmed by fluorescence intensity readings. Hubs of Ncl protein signal were observed in the nuclei of KO EC and pre-HSCs, with the number of the larger hubs enhanced significantly after Atg5 deletion (Fig. 8b–e and Supplementary Fig. 8a–c), implicating a role for Atg5 in the distribution of Ncl in the cell nucleus and cytoplasm.

As mitochondrial activity was affected in the EC and pre-HSC II by Atg5 deletion, by immunostaining displayed the localization of mitochondrial marker Atp5A and Ncl in the EC, pre-HSC I and pre-HSC II single cells. More than 30 single cells were checked and almost all Atp5A signals were localized outside of the nucleus whilst Ncl positive signals were in/out of the nucleus, however, double positive

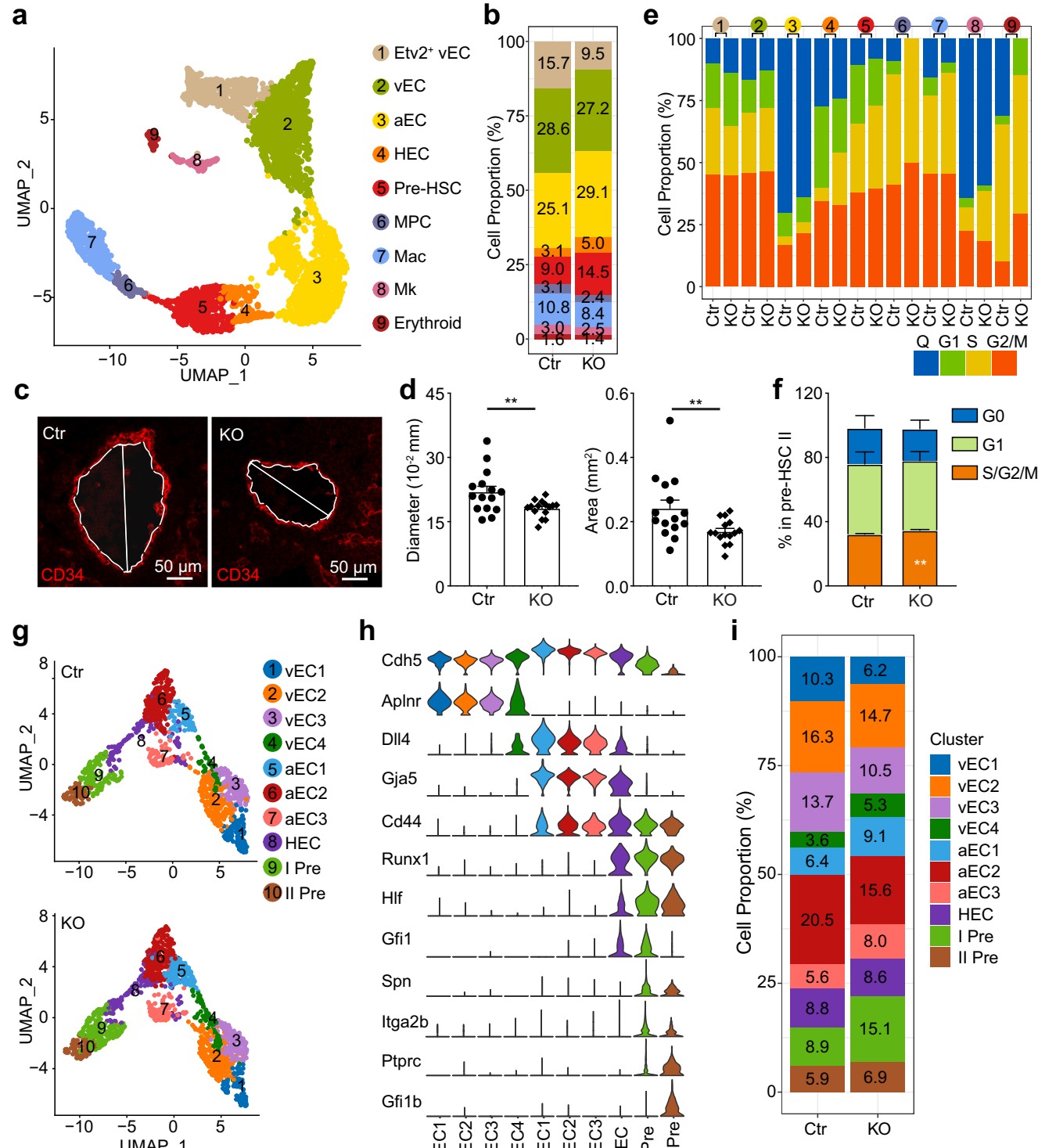

**Fig. 5 | Transcriptomic profiling of hematopoietic related cells in the control and KO AGM region by scRNA-Seq. a** UMAP plots visualized nine clusters from hematopoietic related cells (EC, HC and Mk). **b** Bar graph showing the portions of each sub-cluster (EC and HC) in the control and KO AGM regions. **c** The schematic diagram for detecting the diameter of aorta in the detected sections based on the staining of CD34. Red = CD34. **d** The diameters and areas of aorta in the control and KO group. Error bars represent mean ± SEM. $n = 15$ sections from 3 independent embryos, **$p < 0.01$. Circles = control (Ctr), inverted squares = KO. **e** Bar charts showing the percentage contribution of different cell cycle phases in the control and KO cells. **f** Flow cytometric analysis confirmed the cell cycle status of pre-HSC II in the E11.5 AGM region. Error bars represent mean ± SEM. $n = 4$ biologically independent experiments, **$p = 0.0047$. **g** UMAP plots visualized the ten clusters from vEC, aEC, HEC and pre-HSCs. **h** Violin graphs displaying the key featured genes in each cell cluster. **i** Comparison analysis showing cell proportions of distinct clusters from EC and pre-HSCs between KO and control group. Statistical significance was determined by one side Student's t-test.

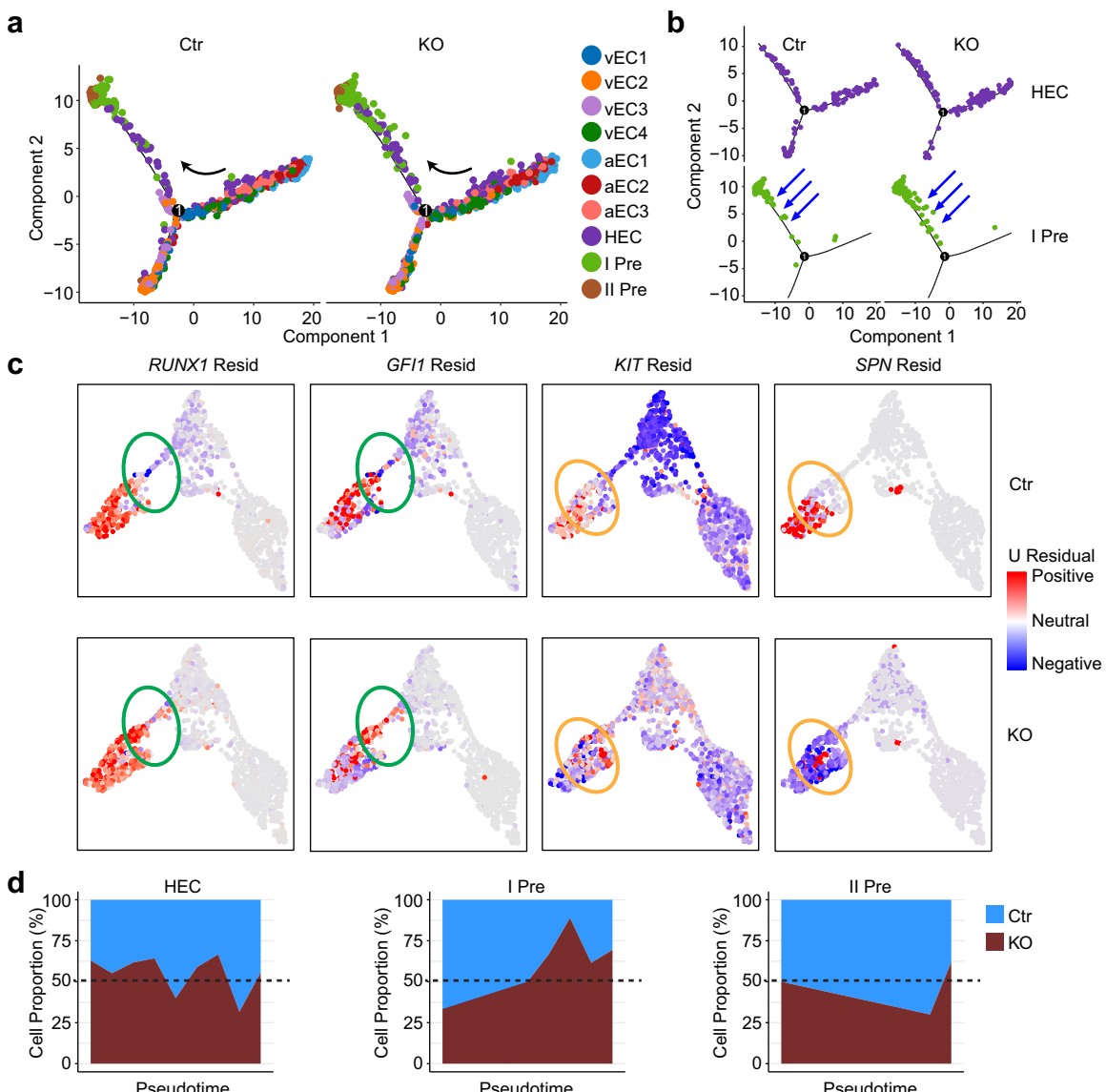

**Fig. 6 | Trajectory analyzing the alteration of endothelial to hematopoietic transition between control and Atg5 deficient AGM region. a** Trajectory analysis by Monocle 2 indicates the alteration of developmental temporal order from EC to pre-HSCs. **b** Trajectory analysis showing the developmental temporal order of HEC and pre-HSC I (I Pre). **c** Comparative RNA velocyto analysis showed the status of unspliced and spliced genes in the control (Ctr) and KO group. The green circle labels the HECs and the orange circle labels pre-HSC Is. **d** Aera plot showing the proportion along the development pseudotime of HEC and pre-HSCs between control and KO embryos.

localization of Ncl+Atp5A was hardly observed, indicating the low possible interaction between mitochondria and Ncl (Supplementary Fig. 8d).

AS1411 (Ncl-binding aptamer) is reported to promote the internalization of Ncl[48]. AS1411 was added in the E10.5 AGM$^{ex}$ culture with/without 3-MA. 3-MA reduced the percentages and cell numbers of HEC and pre-HSC I and II, in line with E11.5 AGM$^{ex}$. Meanwhile, the existence of AS1411 in the 3-MA treatment group enhanced the percentages and cell numbers of HEC and pre-HSC I compared to 3-MA treatment, but not in pre-HSC II (Fig. 8f-i and Supplementary Fig. 1n-o, 8e-g), indicating AS1411 rescues the hematopoietic-related phenotype induced by the inhibition of autophagy. Furthermore, AS1411 failed to rescue the total number of HPC after explant culture, but the potential of BFU-E was recovered partially in the existence of AS1411, even if CFU-GM and CFU-GEMM were not changed (Fig. 8j-m and Supplementary Fig. 8h, i). The fluorescence intensity and hubs of Ncl were not rescued (Supplementary Fig. 8j-l). Since the inhibition ability of 3-MA was very

strong compared with KO mouse model, we modified the experimental setup. 3-MA treatment occurred one day instead of three days. The stronger rescue roles of AS1411 in the number/percentage of PK44 HEC and pre-HSC I$^{plus}$ were observed, although it still failed to play functions in pre-HSC II$^{plus}$ (Fig. 8n–q and Supplementary Fig. 8m). Furthermore, the fluorescence intensity of Ncl was rescued compared with 3-MA group, as well as the number bigger hubs (Supplementary Fig. 8n). Altogether, these data suggest that autophagy plays key role in the EHT through the Ncl pathway.

## Discussion

Here, we have identified that Atg5, an autophagy-related gene, is a regulator of hematopoietic precursor formation/maturation into mature hematopoietic progenitor/stem cells during embryonic development, through blockage of the autophagic process. Mechanistically, Atg5 plays a role in endothelial to hematopoietic transition by the Ncl pathway.

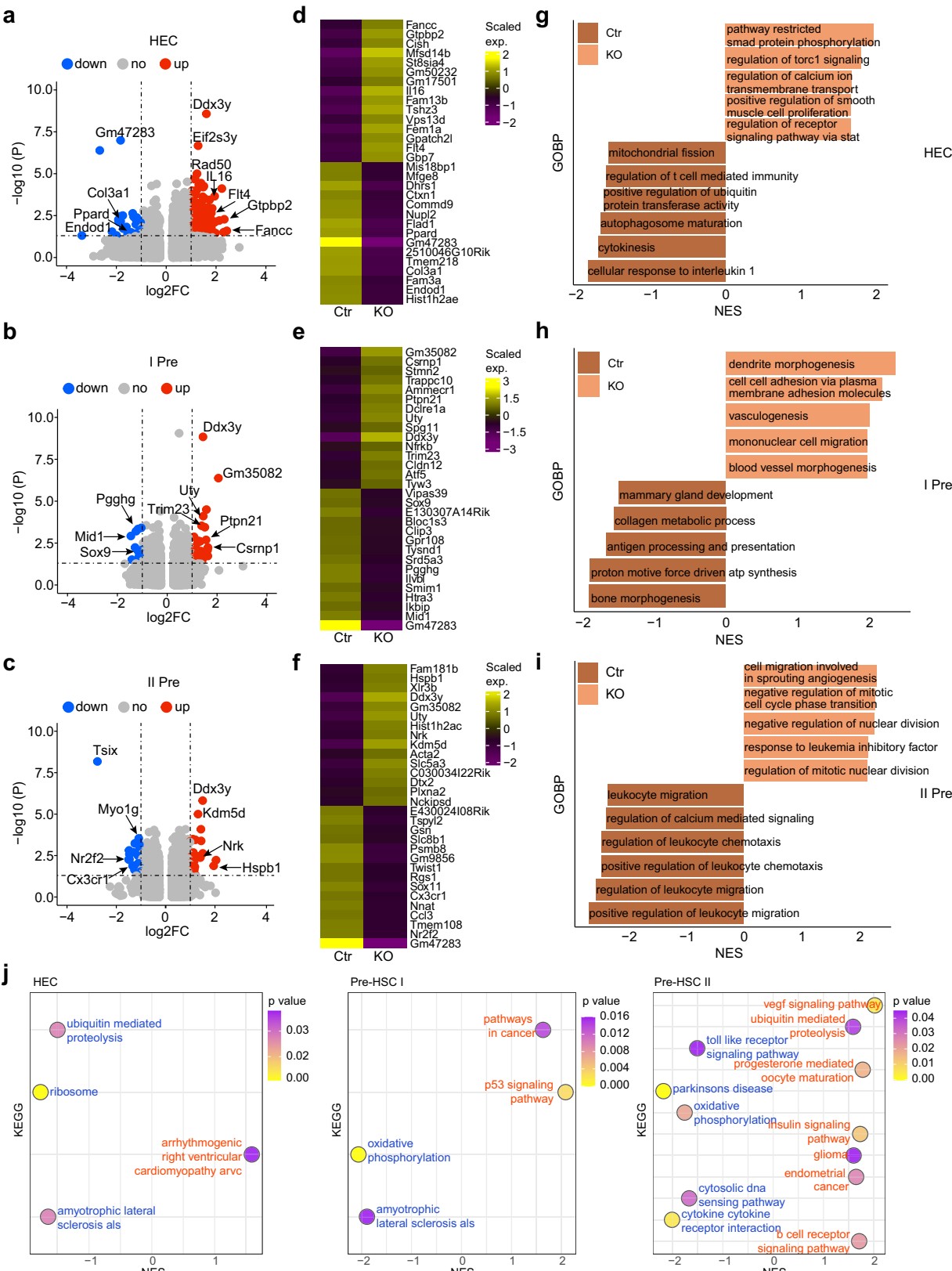

**Fig. 7 | The differences in gene expression and biological process from HEC and pre-HSCs after Atg5 depletion. a–c** Volcano graphs showing the up/down regulated genes in the KO HEC (**a**), pre-HSC I (**b**) and pre-HSC II (**c**) compared to control. Some up or down regulated genes were indicated. **d–f** Top 15 differentially expressed genes (DEGs) in the HEC (**d**), pre-HSC I (**e**) and pre-HSC II (**f**) compared KO with control group by scaled gene expression. **g–i** The enriched genes related to the biological processes are involved in the hematopoietic regulation and autophagy related process in the HEC (**g**), pre-HSC I (**h**) and pre-HSC II (**i**). KO vs control, $p < 0.05$. **j** The normalized enrichment scores (NES) of KEGG pathways by GSEA analysis showing the alteration of ubiquitin-mediated proteolysis, and oxidative phosphorylation in the EC, pre-HSC I and pre-HSC II after Atg5 deletion.

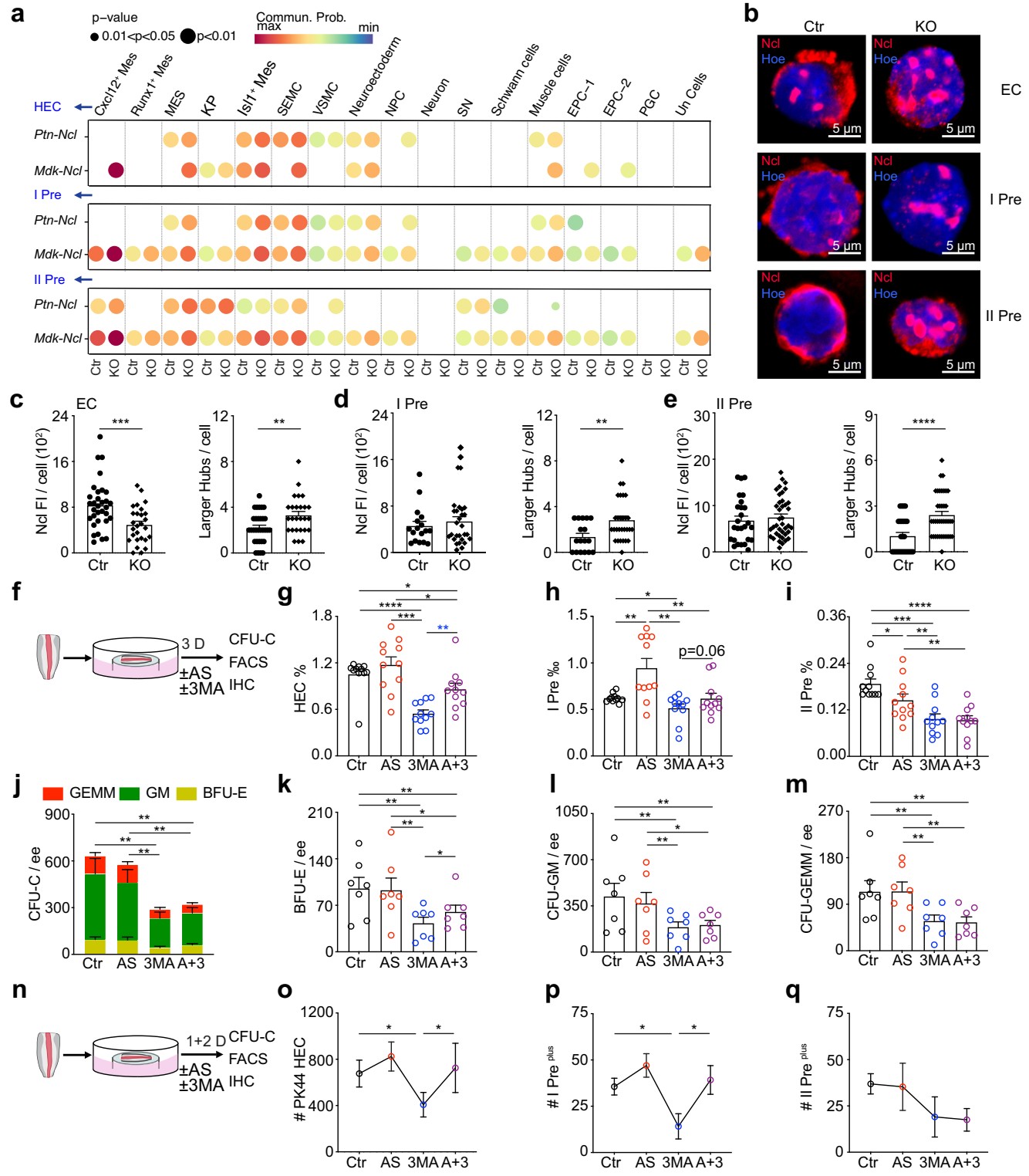

Autophagy is a conserved process, involved in HSC maintenance, survival, and differentiation mediated by different Atgs. Atg5 is the one of 'core' key autophagy components. Atg5 is required for the development of pro- to pre-B cells and for maintaining the B1-a B cells in the periphery[31]. The lack of Atg5 in hematopoietic cells (via Vav-cre mediated deletion) leads to the impaired reconstitution ability of HSCs[30]. Additionally, Atg7 is essential for adult HSC function[32,49], but less essential in fetal liver HSC. In the fetal liver, Fip200 influences the number and frequency of HSC[32]. Our data

illustrate that deficiency of Atg5 in the endothelial cells resulted in the alteration of lineage output from fetal liver HSC by ex vivo culture system, in line with the roles of Atg5 in bone marrow HSC[50]. Meanwhile, Atg5 depletion reduced the number of hematopoietic progenitor cells (CFU-Cs), consistent with the role of Atgs (including Atg5) in myeloid cells of zebrafish[51], and impaired the HSC function by transplantation in the AGM region (pre-liver stage). Therefore, Atg5 plays pivotal roles not only in the fetal liver stage, but also in the earlier stage of hematopoiesis in the embryo, regulating HS/PC

**Fig. 8 | Atg5 regulates endothelial to hematopoietic transition through Ncl pathway. a** Comparison analysis of the expression/interactions (Ptn-Ncl and Mdk-Ncl) between ligands from niche cells (excluding EC, pre-HSCs and Mk) and receptors from HEC, pre-HSC I and pre-HSC II in the control and KO group, showing the enhanced Ptn-Ncl and Mdk-Ncl interplays in these three fractions of KO group compared to control cells. **b** Representative immunostaining data of Ncl in the EC, pre-HSC I (I Pre) and pre-HSC II (II Pre) sorted from E10.5 control (Ctr) and KO AGM regions. Red = Ncl, Blue = Hoe (Hoechst). **c**–**e** Fluorescence intensity (FI) in each cell fraction was calculated by image J and hubs (maximum length of each hub over one micrometer) of Ncl signal in the detected cells. Error bars represent mean ± SEM. Each dot represents a cell. EC ($n = 32$ for control, $n = 27$ for KO), I Pre ($n = 17$ for control, $n = 29$ for KO) and II Pre ($n = 26$ for control, $n = 39$ for KO) were examined over 3 independent experiments). **$p < 0.001$, ***$p = 0.0005$, ****$p < 0.0001$. **f** Experimental set-up for 3 days explant cultures. Adobe Illustrator software was used for drawing this figure. **g**–**i** 3 days E10.5 AGM explant cultures showing the rescue function in the percentages of HEC (**g**) after 3-MA and AS1411 treatment, but not in the pre-HSC I (**h**) and pre-HSC II (**i**) fractions. AS = AS1411, the aptamer of Ncl (also known as AGRO100). 3 MA = 3-MA, the inhibitor of autophagy. A + 3 = AS1411 + 3 MA. Error bars represent mean ± SEM. $n = 11$ embryos over 11 independent experiments, *$p < 0.05$, **$p < 0.01$, ***$p < 0.001$,****$p < 0.0001$. **j** Total CFU-C number per AGM explant and number of each hematopoietic colony type (indicated by color bars) in the existence of AS1411 and 3-MA. Error bars represent mean ± SEM. $n = 7$ embryos examined over 7 independent experiments,**$p < 0.01$. **k**–**m** The number of BFU-E per AGM explant was increased in the AS1411 + 3-MA group compared to 3-MA group, but not in the CFU-GM and CFU-GEMM. Error bars represent mean ± SEM. $n = 7$ embryos over 7 independent experiments, *$p < 0.05$, **$p < 0.01$. **n** Experimental set-up for 1 + 2 days explant cultures. Adobe Illustrator software was used for drawing this figure. **o**–**q** The number of PK44 (CD31⁺CD201⁺cKit⁺CD44⁺CD45⁻CD41⁻) and pre-HSC I$^{plus}$ was rescued by AS1411 in the inhibition of 3-MA, but not in pre-HSC II$^{plus}$. $n = 3$ embryos over 3 independent experiments, *$p < 0.05$. Statistical significance was determined by one side Student's t-test.

development and thus providing support for a role for autophagy in embryonic hematopoiesis.

LC3 (Map1lc3) is a cytosolic protein that, when lipidated, localizes to the surface of autophagosomes as an autophagosome marker. The transgenic mouse model LC3$^{R/G}$, with RFP and EGFP, is used to observe and quantify the status of autophagy in vivo. RFP signal is weaker compared to GFP, which is different from that in the adult bone marrow[39]. Even though RFP⁺GFP⁺, RFP⁺GFP⁻, and RFP⁻GFP⁻ fraction were observed, the RFP signal was not as strong as that in the adult bone marrow since we detected the RFP signals in the bone marrow as control, suggesting the distinct autophagy activity between embryos and adults at steady state, might be due to the distinct protein synthesis. HS/PCs in the fetal liver are mostly LC3-RFP⁺GFP⁺ compared with that in Lin⁻, indicating RFP⁺GFP⁺ signals are related to the immature states. Meanwhile, in the AGM region, RFP⁺GFP⁺ signals are higher in the pre-HSC I than in the pre-HSC II, possibly because pre-HSC II fraction is less enriched and may contain mature myeloid cells. Altogether, the autophagic status varies between the hematopoietic cell fractions.

Atg5 knockout delays the maturation and reduces the survival of adult-generated neurons in the hippocampus[52]. Consistent with neuron development, our data demonstrate that autophagy regulates the maturation of hematopoietic precursors in three aspects. Firstly, our cocultures show that the abilities of maturation in pre-HSC I and II are related to the stage of earlier autophagy (RFP⁺GFP⁺), suggesting that RFP⁺GFP⁺ pre-HSCs represent an immature cell state. Secondly, the percentages and cell numbers of hematopoietic precursors are enhanced in the Atg5 KO AGM region, indicating the blockade of maturation (in vivo data). Lastly, the GFP signal is higher in the hematopoietic precursors of LC3$^{R/G}$-KO AGM, based on the blockade of autophagosome formation, suggesting that Atg5 deletion maintains immature hematopoietic precursors. We couldn't exclude the possible contribution of the enhanced pre-HSC I formation, but the maturation of hematopoietic precursors is indeed regulated by autophagy.

Aberrant autophagy also modulates leukemogenesis. The autophagic regulator of the PI3K/AKT/mTOR pathways has been implicated in leukemogenesis. Atg5 plays roles in MLL-AF9 AML initiation but not in the secondary transplanted leukemia stem cells[53]. In adults, autophagy maintains HSC function and inhibits senescence via Foxo3a and Bag3-dependent manner[39,54]. Lysosomes also regulate HSC metabolism and balance the maintenance of HSC quiescence versus activation[55]. Interestingly, our data show that Atg5 deficiency partially blocks the autophagic process in the hematopoietic-related cell fractions, such as endothelial cells, and pre-HSCs, particularly in hematopoietic precursors. It likely results in the failure of clearance through the lysosome degradation, which possibly contributes to the increase of cell number in hematopoietic clusters. In agreement with this, the transcriptomic data display biologic processes related to

autophagosome maturation and mitochondrial fission/fusion after Atg5 deletion. Recent reports show that mitochondrial membrane potential is a successful approach to separate quiescent HSCs[56]. We have found that the mitochondrial activity (TMRE/MTG, energy metabolism) is decreased in the hematopoietic precursors after Atg5 deletion.

RNA velocity analysis supports that the splicing status of Runx1 and Gfi1 are altered by Atg5 deletion, in line with the key roles of Runx1 and Gfi1 in the EHT process[12,13,15]. Recent report has shown the regulation of Myd88-dependent TLR signaling on hematopoietic clusters formation[57], consistently, our KEGG signaling pathway analysis showed the reduced toll like receptor pathways. Except that, the ubiquitin-mediated proteolysis pathway had different trend in the EC and pre-HSC II, implying the possible dynamic roles of autophagy in protein degradation during EHT stage. Recent report has displayed the involvement of autophagy in the protein hemostasis in the bone marrow HSPC[39]. Rare number of pre-HSCs restricts to check more accurate autophagic flux states and protein degradation of pre-HSCs in the AGM region. However, it remains to be studied whether other autophagy mechanisms, such as mitophagy regulates hematopoiesis and how autophagy affects the metabolism of mitochondrial, protein hemostasis in the embryo.

Hematopoietic precursor cells bud from hemogenic endothelial cells. Recent research shows that the HECs are derived from specified-arterial endothelial cells[9]. We specifically deleted Atg5 in ECs to investigate the role(s) of autophagy. Our scRNA-seq data shows that the development processes of EC are interrupted, in consistent with the reduced aorta size/area. A recent report demonstrates that autophagy modulates EC junctions by repressing the migration of neutrophils[58]. From our bioinformatics analysis, KO HECs are highly enriched in genes restricted to Smad protein phosphorylation, in line with our previous report that smad4 restricts the formation of hematopoietic clusters[17]. Meanwhile, Atg5 deleted pre-HSC I/IIs still sustain the ability for vascular development, with the impairment of migration. Hence, our data illustrate that autophagy likely affects the acquisition of pre-HSC hematopoietic capacity.

It is reported that the Ncl distribution links with the cell cycle of HSC in the bone marrow[37]. Our transcriptomic data have revealed alterations in the cell cycle in Atg5 deleted pre-HSCs. Ncl transcripts are increased in the KO as compared to control ECs, but the protein level is not enhanced. Interestingly, the distribution of Ncl in the intracellular compartments is affected in the KO EC and pre-HSCs, and is likely affecting the hemogenic potential of HECs and the maturation of pre-HSC. The existence of Ncl binding aptamer (AS1411) partially rescues the number and frequency of HEC and pre-HSC I in explant cultures, confirming that Ncl indeed mediates the regulation of autophagy on the hematopoiesis. However, it appears to be irreversible, as autophagy impairs pre-HSC II development by 3-MA (inhibitor

of PI3K pathways and non-specific inhibitor of autophagy), even if only one day 3-MA treatment. 3-MA and chloroquine/ bafilomycin A1 inhibit the different stages of autophagy, but three of them have similar functions on hematopoietic development. In particular, 3-MA exhibits greater effects on hematopoiesis as compared to the effects of Atg5 deletion, possibly explaining its irreversible effects on hematopoietic cells.

In conclusion, we have found that autophagy plays roles in hematopoietic development, particularly in the maturation of hematopoietic precursors into functional HSC, with the modulation on the hemogenic potential of HEC and the maturation of pre-HSC I through Ncl pathways, providing a potential regulator for HS/PC regeneration.

## Methods

### Mouse models
LC3-RFP-EGFP (LC3[R/G], from Jackson lab.[29]), *VE-cadherin-Cre* (from Bing Liu[59]), *Atg5[fl/fl]* (from the RIKEN BioResource Center[60]) mice (8-20 weeks) were used for timed mating and C57BL/6-Ly5.1/1[61] (8-12 weeks) mice were as transplantation recipients.

Mice were housed in the animal facility of Southern Medical University, and mice experiments were approved by the ethics committee of Southern Medical University (L2018273) .

### Mouse embryo generation
LC3[R/G] embryos were generated from LC3[R/G] male mice crossed with C57BL/6-Ly5.2/2 female mice. Male *Vec-Cre;Atg5[fl/+]* mice were crossed with female *Atg5[fl/fl]* mice for Atg5 deficient embryos (Ly5.2/2) (KO: *Vec-Cre;Atg5[fl/fl]*; Control: *Vec-Cre[-];Atg5[fl/+]* or *Vec-Cre[-];Atg5[fl/fl]*). The embryo stages were identified by counting somite pairs and tails were used for genotyping.

### Flow cytometric analysis
Cells from adult bone marrow, fetal liver, and AGM regions were used for flow cytometric analysis. In the adult bone marrow, phenotypically defined HSCs were analyzed by lineage cocktail (Ter119, Gr1, NK1.1, CD3, B220, Mac1), cKit, Sca1, CD150, and CD48. In fetal liver, phenotypic defined HS/PCs (Lin[-]Sca1[+]Mac1[low]) and HSCs (Lin[-]Sca1[+]Mac1[low]CD201[+]) were analyzed by lineage cocktail (Ter119, Gr1, NK1.1, CD3, B220), Sca1, Mac1 and CD201. In the AGM region, digested cells were stained by antibody combination: CD31, CD41, CD45, CD43, CD44, DLL4, CD201, cKit for HECs and pre-HSCs. 7AAD or Hoechst was used for dead cell exclusion (Table S2). For the cell cycle, Ki67 and Dapi were used. For checking mitochondrial activity, cells were washed twice in DBPS after surface marker staining and then resuspended gently in prewarmed (37 °C) DPBS containing the Mito-Tracker® probe (M7514, Invitrogen, 50 nM) and TMRE (ENZ-52309, Enzo, 100 nM). After 15 minutes of incubation at 37 °C by protecting from light, cells were analyzed by flow cytometry following washed in DPBS. Some cells were used for checking gene expression (Table S3).

### Hematopoietic progenitor and stem cell assays
Single-cell suspensions from AGM and fetal liver or cultures were plated in the methylcellulose (M3434; Stem Cell Technologies) for CFU-C assay. The counting and quantification of CFU-C was according to previous report[19]. E11.5 control or KO AGM cells (1-1.5 ee, Ly5.2/2) were co-injected intravenously with supporting cells (2×10[5] leukocytes from bone marrow Ly5.1/1) into 8.5 Gy (4.5 Gy+4 Gy) irradiated recipient mice (Ly5.1/1). Peripheral blood chimerism assays were performed at 4, 12/16 weeks after transplantation. The recipient with chimerism ≥ 5% is considered successful engraftment.

### Explant cultures and OP9-DL1 cocultures
AGM explant (AGM[ex]) culture was performed as previously described[2]. In brief, AGMs were deposited on a nylon membrane (Millipore) placed on metallic supports and cultured in MyeloCult M5300 or H5100

(Stem Cell Technologies) supplemented with 10 μM hydrocortisone (Sigma-Aldrich). After 6 or 12 hours or 1-4 days culture, explants were collected and dissociated into single cells by 0.125% collagenase digestion (Sigma-Aldrich) for flow cytometry analysis and further culture. In some conditions, 3-methyladenine (3-MA, Merck, 1 mM), chloroquine (CQ, Selleck, 2.5 or 5 μM), bafilomycin A1 (Baf, Selleck, 50 or 100 nM), and AS1411 (the aptamer of nucleolin, 5′-GGTGGT GGTGGTTGTGGTGGTGGTGG-3′, 10 μM, from Ruibiotech) were added into the explant cultures.

Endothelial cells, pre-HSC I and pre-HSC II cells were sorted from AGM region and were co-cultured with OP9-DL1 cells (stem cell factor, 100 ng/mL; IL-3, 100 ng/mL; and Flt3-ligand, 100 ng/mL; PeproTech) or OP9 cells (stem cell factor, 20 ng/mL; IL-7, 10 ng/mL; and Flt3-ligand, 10 ng/mL; PeproTech) for 7 days as previous report[62] and cells were harvested by mechanical pipetting for flow cytometric analysis.

### Immunostaining
Immunostaining was performed as described previously[9]. E10.5 embryos were fixed (2% paraformaldehyde/PBS, 20 min, 4 °C), equilibrated in 20% sucrose/PBS at 4 °C overnight and then embed in the Tissue Tek before freezing. From each WT and KO embryo, we sequenced the sections and chose 4-5 sections per embryo from -100 sections at the similar area along the rostral-caudal axis. After additional blocking of endogenous biotin step, antibodies staining were performed. Primary and secondary antibody were used at the following concentrations: Anti-c-Kit (1:100, BD), biotinylated anti-CD34 (1:200, eBioscience), Runx1 (1:500, Abcam) and Hoechst for nuclear staining. Additionally, Ncl (1:100), GFP (MBL, 1:200), RFP (Rockland, 1:200), Lamp1 (Abcam, 1:100) or p62 (Abcam, 1:400) were used for staining ECs and pre-HSCs sorted by flow cytometry and spun onto slides. Secondary antibodies: Alexa Flour 647 anti-rat IgG (1:500), AF488 anti-rabbit IgG (1:1000) and Streptavidin Cy3 (1:1000). The image procedures were performed by confocal microscope (Zeiss LSM 880).

### Single-cell RNA-sequencing
For droplet-based scRNA-seq (10x genomics), AGM cells were sorted by flow cytometry according to Ter119[-] cells or Ter119[-]CD31[+]/CD45[+]/CD41[+] cells, and then the latter was mixed with negative cells (Table S1). Libraries were produced with a Chromium system (10×Genomics, PN1000268) following the manufacture's instruction and sequenced on Illumina Novaseq 6000 platform in 150 bp paired-end manner (sequenced by Novogene and Berry Genomics).

### Single-cell transcriptome analysis and quality control
Sequencing data from 10x Genomics was processed with the Cell Ranger software for each sample. Raw data in FASTQ format was processed and aligned to the mm10 mouse reference genome (https://cf.10xgenomics.com/supp/cell-exp/refdata-gex-mm10-2020-A.tar.gz) with the Cell Ranger v5.0.1 pipeline(10x Genomics, https://www.10xgenom.ics.com/). Doublet cells were removed using DoubletFinder R package (v2.0.3)[63] with the recommended doublet rate by the pipeline. Then filter low-quality cells, we retained cells: (1) expressing 500−7000 genes, (2) less than 10% of reads mapped to mitochondrial genes, and (3) less than 40% of reads mapped to ribosome genes. We sequenced 74,754 single cells, and retained 52,836 cells after the quality-control process of the primary sequencing data.

### Dimensionality reduction and cell clustering analysis
R package Seurat (v4.1.0) was used for downstream analysis and visualization. Raw counts were normalized with the "NormalizeData" function. Highly variable genes (HVGs) were selected by using the "FindVariableFeatures" function with default parameters, which were applied for principal-component analysis (PCA), remove batch effects, data integration and cell clustering.

We removed batch effects from different samples at the same developmental stage by canonical correlation analysis which was conducted in Seurat. Data integration was performed with the "FindIntegrationAnchors" function to determine the integration anchors. Then, based on the union of the top 2,000 HVGs of each dataset, we used the "IntegrateData" function to perform the dataset integration.

We took the union of the top 2000 genes with the highest expression and dispersion from both datasets used for PCA. Then the "ScaleData" function was implemented to scale and center the genes in the dataset. Dimension reduction was conducted using the top significant principal components (PCs) and visualized by UMAP. Finally, we used the "FindClusters" function to identify clusters of cells. Marker genes were identified depending on the adjusted p-values < 0.05 (determined by two-sided Wilcoxon rank-sum test and adjusted using Bonferroni correction) (Supplementary Data 1).

### Identification of DEGs and enrichment analysis

DEGs were identified by "FindMarkers" function in Seurat using the Wilcox method. Genes with the absolute value of log2 fold change >1 and p-value < 0.05 were considered as DEGs (Supplementary Data 1). Then we sequenced these genes from highest to lowest according to log2 fold change in KO to control. Gene Ontology Biological Process(GOBP) Kyoto Encyclopedia of Genes and Genomes (KEGG) enrichment analyses of these lists were conducted using GSEA (v4.2.3, https://www.gsea-msigdb.org/gsea/index.jsp)[64] (Supplementary Data 2).

### Cell cycle analysis

Cell cycle-related genes set of 43 G1/S genes and 54 G2/M genes were used for cell cycle analysis[65,66]. We classified the cycling phases using a similar method to Tirosh et al.[9,67]. We assigned a corresponding cell cycle identity to each cell by calculating the average expression of each gene set. Cells with G1/S score < 2 and G2/M score < 2 were assigned as "quiescent", else as "proliferative". Among proliferative cells: 1) G2/M score > G1/S score, assigned as "G2/M", 2) G1/S score > G2/M score and G2/M score < 2, assigned as "G1", 3) G1/S score > G2/M score and G2/M score ≥ 2, assigned as "S".

### Trajectory analysis

Monocle 2 (version 2.22.0)[68] was used to determine the pseudotime of ten subclusters from vEC, aEC, and pre-HSC clusters, and pseudotime was scaled from 0 to 1. Identified significant genes used in pseudotime analysis by the "dispersionTable" function. We followed the official vignette with recommended parameters.

### RNA velocity

To predict the direction of EHT from EC, HEC and pre-HSC, we used the velocyto (v0.17.17)[69] with default parameters. Further analysis and visualization were carried out using R package velocyto.R (v0.6).

### Cell-cell communication analysis

To identify and visualize the communicating interactions between HEC/pre-HSCs and other cells (excluding EC and HCs) at a single-cell resolution by using R package CellChat (version 1.1.3)[70]. We followed the official workflow to compare the potential ligand-receptor interactions between control and KO groups were obtained.

### QUANTIFICATION AND STATISTICAL ANALYSIS

All graphs were generated using GraphPad Prism 8. All data are presented as the mean ± SEM. One side Student t-test is for comparisons of 2 groups except for immunostaining data. And one-way ANOVA analysis of variance test is for comparisons of >2 groups. Mann-Whitney U-test is for comparison of hematopoietic cluster cells by immunostaining. $p < 0.05$ was considered significant. $*p < 0.05$, $**p < 0.01$, $***p < 0.001$, $****p < 0.0001$. Further statistical details of experiments can be found in the figure legends. The number of biological replicates is indicated with 'n'.

### Reporting summary

Further information on research design is available in the Nature Portfolio Reporting Summary linked to this article.

## Data availability

The raw sequencing data generated in the present study are deposited in the GSA database (Genome Sequence Archive in BIG Data Center, Beijing Institute of Genomics, Chinese Academy of Sciences) with the accession number CRA008812. The processed datasets generated by this study are deposited in the OMIX database (China National Center for Bioinformation/Beijing Institute of Genomics, Chinese Academy of Sciences), under accession OMIX002331. All other data are provided in the Supplementary Information/Source Data file. Source data are provided with this paper.

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

## Acknowledgements

The authors thank all lab members for helpful comments. We also thank Wenyue Xu for for the contribution of Atg5$^{fl/fl}$ and Bing Liu for Vec-Cre mice. We thank Dr. B. Liu and Dr. Y. Lan for critical discussion. The authors thank the Institute of Hematology, Jinan University, for providing flow cytometric sorting. This work was supported by grants from the National Key Research and Development Program (2019YFA0801802, 2019YFA0111100), the National Natural Science Foundation of China (82070105).

## Author contributions

Z.L. conceived the project and designed the experiments. Y.L., L.S., S.L., and Y.C. performed experiments. Y.C. mainly performed bioinformatics analysis. H.C. and W.L. irradiated mice. Z.Y. helped with genotyping. J.L. helped with bioinformatics analysis. X.L., Z.C., and B.Y. helped for FACS. Z.L. and Y.L. wrote the manuscript, and Elaine Dzierzak revised the manuscript.

## Competing interests

The authors declare no competing interests.
