## [Peer Review File · Nature Communications]

Autophagy regulates the maturation of hematopoietic precursors in the embryoReviewer #1 (Remarks to the Author):

In this manuscript, Liu and colleagues investigate the role of autophagy in early embryonic haematopoietic stem cell development. Although autophagy has been extensively studied in HSC biology, there is little on the very earliest stages of embryonic development, so there is definitely novelty here.

My major concerns are around the interpretation of data from the LC3-RFP-GFP mouse model. This mouse strain has recently been published in Chua et al. (Cell Stem Cell 2023), looking at adult HSCs, and in this manuscript the authors are able to reproduce the expression pattern and gating strategy of RFP/GFP. However, in fetal liver HSCs the pattern is quite different, and the control and GFP-RFP- populations are not distinguishable, unlike adult HSCs. The description of GFP-RFP- as indicating autolysosomes (line 128) is incorrect; they would be GFP-RFP+. For their analysis, they have selected a different gating strategy than in the adult HSCs, which seems strange. Showing images (S1G-H) which show GFP and RFP expression does not resolve this. The data as shown simply reflects a continuum of expression of both proteins and so cannot allow much comment to be made about autophagy. Since LC3-GFP-RFP is expressed under a CAG promoter in this mouse, GFP+RFP+ does not reflect autophagosome-bound LC3 and does not indicate active autophagy. As a minimum, the authors need to use bafilomycin or similar to show changes in flux.

Related to Fig 2

-The authors need to confirm deletion of Atg5.
-Line 210 (Fig. 2I-J). There is no statistical test here, and the authors are using an inappropriate measurement of the central tendency of the data in the text which clearly doesn't reflect the actual results. The claim in line 212 'These data show the critical function of autophagy in the HS/PC function in the AGM region' is not substantiated.

Related to Fig 3

3K, M - the data is clearly non-normally distributed and the choice of statistical test is inappropriate.

Related to Fig 4

-This series of experiments is generally incorrectly interpreted. Since Atg5 deletion will affect LC3 lipidation and therefore autophagy, it will prevent incorporation of overexpressed LC3 into autophagosomes, and lead to their accumulation, which is shown in 4E for GFP. However, RFP is unchanged and this suggests that there is a significant issue with its use in this mouse.

-The total Mitotracker Green MFI needs to be shown as well as the ratio (Fig S4P-S). Since MTG and TMRE clash with GFP/RFP in flow cytometry, I assume that WT mice were used for these experiments, but that is not stated in the figure legend or methods.

Related to Fig 6

- I very much struggle to see that there is a difference in 6A between conditions in the pseudotime distribution.
- 6F-L - why are Ctr and KO different colors here? The description of this data is confusing as the comparator needs to be constant (i.e. always compare KO vs WT rather than switching. There is no mention of the P values for the data.

Related to Fig 7

- 7F-Q; 3-MA is a highly non-specific inhibitor of autophagy and should not be used if a genetic deletion model is available, or if it is, its non-specificity needs to be specifically highlighted. It is particularly incorrectly described in line 527 of the discussion.

In general, there no mention of the times an experiment was independently replicated.

Reviewer #2 (Remarks to the Author):

The work of Liu et al is an original report demonstrating the involvement of autophagy in the first stages of hematopoietic differentiation. The authors elegantly show that the amount of autophagic activity correlates with different immature stages of HSC development. They demonstrate that Atg5 is required for a proper HSC development from endothelial cells. The authors widely characterize the embryonic hematopoietic AGM cells by scRNAseq and demonstrate alterations in some cell types. Finally, they can show that Ncl pathway is downstream of Autophagy to regulate HSC development.

This is a novel work and it adds important knowledge to the field. The experiments are well designed and the figures are of high quality, however the story is not presented in a linear way and it is somehow difficult to follow. The single cell RNA data information should be selected and only relevant figures should be shown in the main figures, there is an overwhelming amount of information, and several figures are not necessary. Instead, other figures showing functional analysis of important characteristics are in supplementary figures. I think that focusing more in the important results would make the impact much stronger.

The paper is now divided in:

- 1- Autophagy is involved in FL and AGM by LC3-reporter mice (Fig 1)
- 2-Atg5 regulates HSPC in the embryo in FL and AGM (Fig 2)
- 3-Formation of HP is altered in the absence of Atg5 AGM (Fig 3)
- 4-Inhibition of autophagy undermines the fusion of autophagosome-lysosome in HP (Fig 4)
- 5-Transcriptomic of Atg5 control/ KO (Fig 5)
- 6-comparative from endothelial to HSC of wt/KO (Fig 5-6)
- 7-Atg5 depletion changes the hematopoietic related biological process (Fig 6)
- 8-Atg influences EHT through nucleolin (Fig 7)

The authors should consider combining results for Fig2 and Fig3, making a more simpler figure in which the results of HSC development were shown. They can probably summarise the data of FL in one figure and dedicate the rest to AGM. Similarly, the analysis on the cluster cells is nice but the differences are not that clear, which also is a little disruptive to the story line. It would have been nice to see the LC3 reporter in AGM sections, or they could show the immunostaining of pre-HSCs from FigS1 in the main figures.

Similarly for Fig 5 and Fig 6, results have to be selected which ones are relevant for the story. Instead, mitochondrial activity from Fig S4P-S could be moved to main figures. MitoTracker Green (MTG) probe and mitochondrial activity experiments are important information, however the authors should speculate why differences are not seen in pre-HSC I when this is supposed to be the population most affected by deletion of Atg5. Moreover, the sentence: 'GeoMFI ratios were reduced dramatically in the EC, pre-HSC II...' is an overinterpretation.

Cell cycle analysis is very important in the EHT to HSC developmental process and should be further explored within the different AGM cell types. Can the authors confirm the cell cycle differences observed by scRNA seq data by Ki67 or other technique?

In Figure 5D, the authors claim that it shows: 'the percentage of EC/HC was increased in the Atg5-deleted group compared to control group', however I cannot appreciate this difference and the figure is not informative in this sense.

In Figure 5G and H, the authors claim that it shows how 'Atg5 deletion leads to the accumulation of pre-HSC I and a delay in the developmental process of pre-HSC I relative to the maturation of pre-HSCs'. However, this is not clear from the figure and authors should highlight the differences in the figure and better explain in the text.

Finally the nucleolin data is relevant and adds a mechanistic component to the work, but require further explanation in the text. They should include this information in the abstract, but also in the introduction (or briefly in results), some background of the pathway and its relationship with autophagy should be included. They include some information in the discussion, but it should be introduced before.

Dear Reviewers,

Thanks a lot for your critical comments. We have addressed them carefully and the manuscript is more concise. We will answer all the points one by one. Please check these details in the following pages.

Best regards,

Zhuan

REVIEWER COMMENTS

Reviewer #1 (Remarks to the Author):

In this manuscript, Liu and colleagues investigate the role of autophagy in early embryonic haematopoietic stem cell development. Although autophagy has been extensively studied in HSC biology, there is little on the very earliest stages of embryonic development, so there is definitely novelty here.

My major concerns are around the interpretation of data from the LC3-RFP-GFP mouse model. This mouse strain has recently been published in Chua et al. (Cell Stem Cell 2023), looking at adult HSCs, and in this manuscript the authors are able to reproduce the expression pattern and gating strategy of RFP/GFP. However, in fetal liver HSCs the pattern is quite different, and the control and GFP-RFP- populations are not distinguishable, unlike adult HSCs. The description of GFP-RFP- as indicating autolysosomes (line 128) is incorrect; they would be GFP-RFP+. For their analysis, they have selected a different gating strategy than in the adult HSCs, which seems strange. Showing images (S1G-H) which show GFP and RFP expression does not resolve this.

Re: Thanks a lot for your points. You are right, we have used different gating strategies between bone marrow and embryonic fetal liver/AGM regions. For bone marrow and fetal liver cell assay, similar conditions or equipment parameters (including voltage and flow rates) were used in the flow cytometric analysis. We have tried to use the same gating strategy as bone marrow, no RFP⁺GFP⁻ cells were observed in the fetal liver (in the following figure a, these figures are not shown in the figures or supplementary figures, because we have shown the gating strategies of BM and fetal liver). GFP and RFP signals appear inconsistent patterns in the distinct organs (bone marrow and fetal liver). Distinct protein synthesis is reported in the fetal liver HSCs compared to bone marrow and autophagy is one of the ways to degrade the mis/unfold protein (Stem Cell Reports, 2021; Current Opinion, 2020), so it makes sense that autophagy status is quite different in the fetal liver and bone marrow. Altogether, that is why we keep using different strategies between bone marrow and fetal liver/AGM region.

For line 128, do you mean that RFP^{low}GFP^{low} represents autolysosome? Based on the flow data from bone marrow, the GFP signal looks lower in the RFP⁺GFP⁻ HSCs compared with GFP⁺RFP⁺ HSCs and RFP signals also appear a bit lower. To confirm this phenomenon, the GeoMFI of

GFP or RFP was calculated in these three fractions (GFP⁺RFP⁺, RFP⁺GFP⁻, RFP⁻GFP⁻ HSCs). GeoMFI of GFP was much lower in RFP⁺GFP⁻ HSCs compared with GFP⁺RFP⁺ fraction in the bone marrow HSCs, the same trend to GeoMFI of RFP (the following figure b), indicating a bit of alteration of RFP in the formation of autolysosome.

In the embryonic AGM cells, flow analysis displayed the level of GFP and RFP signals is obviously lower in RFP^{low}GFP^{low} cells than in the RFP⁺GFP⁺ fractions (figure 1b and 1d). Meanwhile, GFP and RFP signals were confirmed by confocal microscopy in the fractions of RFP⁺GFP⁺, RFP^{low}GFP^{low}, RFP⁻GFP⁻ endothelial cells, pre-HSC I and II. As is expected the signals of GFP (GeoMFI) in RFP^{low}GFP^{low} are much lower than that of RFP⁺GFP⁺ ECs, a similar trend to RFP (the following figure c-d, figure 1f and supplementary figure 1g).

Altogether with bone marrow data, GFP^{low}RFP^{low} cells in AGM region are more similar to RFP⁺GFP⁻ cells of bone marrow, representing the autolysosome. Based on the gating strategy, it is more accurate to name this fraction as GFP^{low}RFP^{low} in our study.

The data as shown simply reflects a continuum of expression of both proteins and so cannot allow much comment to be made about autophagy. Since LC3-GFP-RFP is expressed under a CAG promoter in this mouse, GFP⁺RFP⁺ does not reflect autophagosome-bound LC3 and does not

indicate active autophagy. As a minimum, the authors need to use bafilomycin or similar to show changes in flux.

Re: Thank you very much for your wonderful comments. Yes, we agree with you that LC3-GFP-RFP is controlled under a CAG promoter in the mouse. In mammalian cells, because the different pH-values exist between autophagosome and lysosome (Li et al., JASN, 2014), there are acidic environment in the lysosome, pH=4-5. The sensitivity between GFP and RFP is distinct, GFP pKa=5.9 and RFP pKa=4.5. So, the fluorescence of GFP is quenched after the fusion of autophagosome with lysosome, while that of RFP is relatively stable and retained even within lysosomes. Based on this, LC3-GFP-RFP is able to report the process of fusion autophagosome and lysosome somehow.

Meanwhile, according to your suggestion, we used chloroquine (CQ) and bafilomycin A1 (Baf1) to inhibit the fusion of autophagosome with lysosome on AGM explant cultures. Because the dosage of CQ or Baf1 is related to the cell line in most published reports, we have tested the effects of CQ or Baf1 on AGM explant cultures based on cell viability. For 12-hour explant cultures, our data showed that the existence of CQ (5 μ M, not 2.5 μ M) decreased the percentages and cell numbers of pre-HSC I and pre-HSC II, but not in the endothelial cells (the following figure a-b, supplementary figure 1p-q). Consistently, Baf1 (V-ATPase inhibitor, 100 nM, not 50 nM) also resulted in the reduction of pre-HSC I and II percentages and cell numbers, (the following figure c-d, supplementary figure 1r-s). Both Baf1 and CQ had similar effects to 3-MA on AGM explant cultures.

Additionally, because the treatment of Baf1 (100 nM, 12h) led to a dramatic reduction in pre-HSC I and II, rare cells are able to gain for analysis, particularly for checking GFP/RFP fluorescence signals. Then, we also tried 6-hour explant cultures, the reduction was only found in the pre-HSC I cells (the following figure e-f, supplementary figure 1t-u), indicating higher sensitivity to autophagic inhibitor in pre-HSC I. Meanwhile, the ratios of GFP/RFP were increased in EC, pre-HSC I, pre-HSC II and hematopoietic clusters after 6-hour Baf1 treatment (the following figure g, supplementary figure 4j). Consistently, the percentage of RFP⁺GFP⁺ cells was increased in the 44⁺ ECs, pre-HSC I (the following figure h, supplementary figure 4k), in line with the effects on Atg5 deletion (LC3^{R/G}-KO), indicating block fusion of autophagosome and lysosome. Taken together, these data suggest autophagy is involved in the hematopoiesis and LC3-RFP-EGFP reporter mouse model is able to label the distinct autophagic status of hematopoietic cells.

Related to Fig 2

-The authors need to confirm deletion of Atg5.

Re: Thanks a lot for your comment. We have shown the deletion efficiency of Atg5 in the supplementary figure 7 of last version. In this revised manuscript, we have moved this figure into supplementary figure 2. Please see the details.

-Line 210 (Fig. 2I-J). There is no statistical test here, and the authors are using an inappropriate measurement of the central tendency of the data in the text which clearly doesn't reflect the actual results. The claim in line 212 'These data show the critical function of autophagy in the HS/PC function in the AGM region' is not substantiated.

Re: Thank you very much for this point. Approximately one adult-repopulated HSCs were observed in the AGM region (Medvinsky et al., *Development*, 2011), and in direct transplantation experiments, normally around 60% of recipients were engrafted even if two embryo equivalent AGM cells were transplanted. According to the methods we have described, 5% of the chimerism is the baseline for positive engrafted recipients. From our transplantation data, there are no successful engrafted recipients in the KO group (0/3), whilst 3 out of 5 recipients were engrafted with $\geq 5\%$ chimerism. Although no statistics were found, Atg5 deletion altered the engrafted ability. In addition, we have modified the text into "the effects of autophagy in the HS/PC activity in the AGM region".

Related to Fig 3

3K, M - the data is clearly non-normally distributed and the choice of statistical test is inappropriate.

Re: Thanks a lot for this point. We have reanalyzed these data by using the Mann-Whitney U-test and corrected the related figures and figure legends. Meanwhile, the details were updated in the methods.

Related to Fig 4

-This series of experiments is generally incorrectly interpreted. Since Atg5 deletion will affect LC3 lipidation and therefore autophagy, it will prevent incorporation of overexpressed LC3 into autophagosomes, and lead to their accumulation, which is shown in 4E for GFP. However, RFP is unchanged and this suggests that there is a significant issue with its use in this mouse.

Re: Thanks a lot for your question. As we mentioned in your comment #2, in mammalian cells, because the different pH-values exist between autophagosome and lysosome (Li et al., JASN, 2014), there are acidic environment in the lysosome, pH=4-5. The sensitivity between GFP and RFP is distinct, GFP pKa=5.9 and RFP pKa=4.5. So, the fluorescence of GFP is quenched after the fusion of autophagosome with lysosome, while that of RFP is relatively stable and retained even within lysosomes. Based on this, LC3-GFP-RFP is able to report the process of fusion autophagosome and lysosome somehow. So, we can test the percentage of GFP⁺RFP⁺ or the signal of GFP or RFP to check the autophagic status in the control or KO mouse model.

From our LC3^{R/G}-KO mouse model, Atg5 deletion blocked the process fusion of autophagosome with the lysosome, which is reported by the GFP fluorescence reduction in the acidic environment of lysosome and the RFP is relatively stable. Meanwhile, the ex vivo experiments with Baf1 treatment (inhibiting autolysosome formation) displayed similar results to LC3^{R/G}-KO AGM cells, suggesting the block fusion of autophagosome with lysosome, with increase of GFP signals. That is why we keep these analyses in the figure 4 and supplementary figure 4.

-The total Mitotracker Green MFI needs to be shown as well as the ratio (Fig S4P-S). Since MTG and TMRE clash with GFP/RFP in flow cytometry, I assume that WT mice were used for these experiments, but that is not stated in the figure legend or methods.

Re: Thank you very much for these comments. We have shown the MFI of MTG and TMRE from control and KO AGM cells (in the following figure a-b, supplementary figure 4l-m). Control group is equal to the WT mice (Vec-Cre⁻;Atg5^{fl/+}, or Vec-Cre⁻;Atg5^{+/+}) you mentioned. The details were added to the methods.

Related to Fig 6

- I very much struggle to see that there is a difference in 6A between conditions in the pseudotime distribution.

Re: Thanks a lot for this point. We have added the shadow to highlight the key parts for checking the differences in Figure 6d of this revised manuscript.

- 6F-L – why are Ctr and KO different colors here? The description of this data is confusing as the comparator needs to be constant (i.e. always compare KO vs WT rather than switching. There is no mention of the P values for the data.

Re: Thanks a lot for your comment. It is a really good point and we have changed them into the same colors. The dark brown presents the process enriched in the control group and the light brown presents that in the KO group. The p-value has been added to the methods.

Related to Fig 7

- 7F-Q; 3-MA is a highly non-specific inhibitor of autophagy and should not be used if a genetic deletion model is available, or if it is, its non-specificity needs to be specifically highlighted. It is particularly incorrectly described in line 527 of the discussion.

Re: Thank you very much for your suggestion. Yes, 3-MA is not a specific inhibitor of autophagy. As is shown in the paper (Orsini et al., 2018, Biochemical pharmacology) that 3-MA is the inhibitor of PI3K pathways, which is involved in the different steps of the autophagy process, especially during the initiation step. Because we couldn't gain 4 embryos (2 control and 2 ko embryos with the same stage) from one litter embryo, and our data showed 3-MA affected HPC and pre-HSC development. So, we used 3-MA treatment in AGM explant for checking the NCL function in autophagy instead of Atg5 KO embryos. Additionally, according to your suggestion related to figure 1, CQ and Baf1 treatment experiments showed reduced effects on the pre-HSC I and II, which is similar to 3-MA treatment. We have also modified the point in line 527 of discussion.

In general, there no mention of the times an experiment was independently replicated.

Re: Thanks a lot for your comment. As we mentioned in the methods, the number of biological replicates is indicated with 'n'. We have added all the experiment times in figure legends.

Reviewer #2 (Remarks to the Author):

The work of Liu et al is an original report demonstrating the involvement of autophagy in the first stages of hematopoietic differentiation. The authors elegantly show that the amount of autophagic activity correlates with different immature stages of HSC development. They demonstrate that Atg5 is required for a proper HSC development from endothelial cells. The authors widely characterize the embryonic hematopoietic AGM cells by scRNAseq and demonstrate alterations in some cell types. Finally, they can show that Ncl pathway is downstream of Autophagy to regulate HSC development. This is a novel work and it adds important knowledge to the field. The experiments are well designed and the figures are of high quality, however the story is not presented in a linear way and it is somehow difficult to follow. The single cell RNA data information should be selected and only relevant figures should be shown in the main figures, there is an overwhelming amount of information, and several figures are not necessary. Instead, other figures showing functional analysis of important characteristics are in supplementary figures. I think that focusing more in the important results would make the impact much stronger.

The paper is now divided in:

- 1- Autophagy is involved in FL and AGM by LC3-reporter mice (Fig 1)
- 2-Atg5 regulates HSPC in the embryo in FL and AGM (Fig 2)
- 3-Formation of HP is altered in the absence of Atg5 AGM (Fig 3)
- 4-Inhibition of autophagy undermines the fusion of autophagosome-lysosome in HP (Fig 4)
- 5-Transcriptomic of Atg5 control/ KO (Fig 5)
- 6-comparative from endothelial to HSC of wt/KO (Fig 5-6)
- 7-Atg5 depletion changes the hematopoietic related biological process (Fig 6)
- 8-Atg influences EHT through nucleolin (Fig 7)

The authors should consider combining results for Fig2 and Fig3, making a more simpler figure in which the results of HSC development were shown. They can probably summarise the data of FL in one figure and dedicate the rest to AGM. Similarly, the analysis on the cluster cells is nice but the differences are not that clear, which also is a little disruptive to the story line.

Re: Thank you very much for your kind comments. We have reorganized the paper according to your suggestion. Please see the details in the revised manuscript. Mainly, we divided all figures into 8 figures and separated the data related to FL and AGM regions.

It would have been nice to see the LC3 reporter in AGM sections, or they could show the immunostaining of pre-HSCs from FigS1 in the main figures.

Re: Thank you very much for this point. We have tried to perform the immunofluorescence staining on AGM LC3^{R/G} sections. However, the clear signals of RFP and GFP were not observed based on the cryosection although we tried different ways to fix tissues (do cryosection after 4% PFA fixation or do fixation by methanol after sections). It might due to the fragile embryonic cells and inappropriate pH-values from the staining buffer. So, we have to change the ways. We sorted live GFP⁺RFP⁺, GFP^{low}RFP^{low}, GFP⁻RFP⁻ ECs, pre-HSC I and II for checking the signals of GFP and RFP under confocal microscopy instead of cryosection immunostaining. The clear GFP and RFP signals were investigated in the RFP⁺GFP⁺ EC, pre-HSC I and pre-HSC II cells, but not in the RFP⁻GFP⁻ ECs (in the following figures, figure 1f and supplementary figure 1g).

Similarly for Fig 5 and Fig 6, results have to be selected which ones are relevant for the story.

Re: Thanks a lot for your comments. We have reorganized the manuscript according to your suggestion as we mentioned, please see the details in the revised manuscript.

Instead, mitochondrial activity from Fig S4P-S could be moved to main figures. MitoTracker Green (MTG) probe and mitochondrial activity experiments are important information, however the authors should speculate why differences are not seen in pre-HSC I when this is supposed to be the population most affected by deletion of Atg5. Moreover, the sentence: 'GeoMFI ratios were reduced dramatically in the EC, pre-HSC II...' is an overinterpretation.

Re: Thanks a lot for your comments. Firstly, the FigS4P-S was moved into Fig 4 (please see the figure 4f). Secondly, no differences about TMRE/MTG were found the fraction of pre-HSC I, but not in other cell populations. Even if we have repeated this experiment 7 times, possibly because the rare cells of pre-HSC I were gained and the mitochondrial heterogeneity of pre-HSC I worsened the variation of TMRE/MTG ratio. Finally, we have modified the sentence you mentioned, please see the correction in the revised manuscript.

Cell cycle analysis is very important in the EHT to HSC developmental process and should be further explored within the different AGM cell types. Can the authors confirm the cell cycle differences observed by scRNA seq data by Ki67 or other technique?

Re: Thanks a lot for your comments. We have checked the cell cycle activity of EC, pre-HSC I and pre-HSC II cells by staining Ki67 and DAPI. In EC and pre-HSC I cells, Atg5 deletion (KO) didn't alter the status of cell cycle, but the percentage of S/G2/M was increased in the pre-HSC II cells (in the following figure, figure 5f and supplementary figure 5g), partially similar to the transcriptomic analysis data with active cell cycle of pre-HSCs in the KO group (figure 5e), possibly due to the different stages of embryos and the distinct genes to analyzing cell cycle between flow and transcriptomic analysis .

In Figure 5D, the authors claim that it shows: 'the percentage of EC/HC was increased in the Atg5-deleted group compared to control group', however I cannot appreciate this difference and the figure is not informative in this sense.

Re: Thanks a lot for your comments. We have corrected this information, which is related to figure 5i in the revised manuscript.

In Figure 5G and H, the authors claim that it shows how 'Atg5 deletion leads to the accumulation of pre-HSC I and a delay in the developmental process of pre-HSC I relative to the maturation of pre-HSCs'. However, this is not clear from the figure and authors should highlight the differences in the figure and better explain in the text.

Re: Thanks a lot for your comments. We have highlighted the key points of pre-HSC I accumulation in the KO group by blue arrows (figure 6b in the modified version) and explained them carefully. Please see the details in the manuscript.

Finally the nucleolin data is relevant and adds a mechanistic component to the work, but require further

explanation in the text. They should include this information in the abstract, but also in the introduction (or briefly in results), some background of the pathway and its relationship with autophagy should be included. They include some information in the discussion, but it should be introduced before.

Re: Thanks a lot for your comments. We have added information related to nucleolin in the revised manuscript according to your suggestion, please see the paragraph 5 of introduction.

Reviewer #1 (Remarks to the Author):

In this revised version of the manuscript, the authors have sadly missed almost every one of my points, which remain unaddressed.

I'm afraid I'm not satisfied with the response to my first point, which perhaps the authors have misunderstood. Their gating is flat out wrong for the LC3-GFP/RFP reporter.

In Fig 1b, the gating is different to S1a. This is not reflective of the relative distribution of the RFP/GFP signal, which seems about the same between HSC and FL. In S1a, the authors have drawn gates which bisect the main population of cells longitudinally, whereas in Fig 1b, the gates transect the cell population. Specifically, in S1a an RFP+GFP- gate is drawn, but this is not done in Fig 1b. The reason this is important is that this represents autolysosomes. In the rebuttal figure 'a', it is not necessary to use the same GFP/RFP signal as seen in HSC to define the gates, but they can be drawn using the same geometry (pattern) as the HSCs. The authors need to reanalyse their data with the same gating geometry as the Chua et al. 2023 paper, and as they have done in S1a.

My comment about line 128 is that GFP-RFP- are not obviously autolysosomes; if the authors measure the RFP/GFP MFI they will be low (rebuttal fig b) as they have been gated as low! The definition of autolysosomes consistent with the literature is RFP+GFP-, which a population gated in S1a (and in the Chua et al paper). To resolve this, the authors need to also stain for Lamp1 (as they have done in 1g in a bulk population) in the cell populations shown in rebuttal fig 'c'.

Unfortunately there are also issues around the use of the bafilomycin experiments. Firstly, the duration of culture with bafilomycin is too long and it's not surprising there was toxicity. Bafilomycin should be used for a short time (2-4h). Secondly, reporting changes in the GFP/RFP ratio is uninformative as bafilomycin will raise lysosomal pH and therefore increase GFP signal notwithstanding any effect on autophagy.

The authors have also failed to understand the issue with using Atg5-LC3-GFP-RFP mice, and copy pasting an irrelevant answer about pH from an earlier section in the response to my comments about figure 4 is not satisfactory. Atg5 deletion does NOT block autophagosome lysosome fusion, it prevents autophagosome elongation by LC3 lipidation. They also do not understand my point about MTG and TMRE clashing with GFP/RFP and do not clarify that the mice lack LC3-GFP-RFP.

In relation to my comments on pseudotime analysis, again there is no visibly discernible difference between WT and KO, and if the authors wish to claim there is, there needs to be some kind of statistical test done.

Reviewer #2 (Remarks to the Author):

The authors have addressed the raised concerns.

Dear Reviewer and Editor,

Thanks a lot for your kind consideration and critical comments. I sincerely apologize that we have misunderstood the points you mentioned in the last version. In this version, we have addressed carefully all the points one by one. Thank you very much and sorry again for the unsatisfied addressment.

Sincerely and best regards,

Zhuan

REVIEWER COMMENTS

Reviewer #1 (Remarks to the Author):

In this revised version of the manuscript, the authors have sadly missed almost every one of my points, which remain unaddressed.

I'm afraid I'm not satisfied with the response to my first point, which perhaps the authors have misunderstood. Their gating is flat out wrong for the LC3-GFP/RFP reporter.

In Fig 1b, the gating is different to S1a. This is not reflective of the relative distribution of the RFP/GFP signal, which seems about the same between HSC and FL. In S1a, the authors have drawn gates which bisect the main population of cells longitudinally, whereas in Fig 1b, the gates transect the cell population. Specifically, in S1a an RFP+GFP- gate is drawn, but this is not done in Fig 1b. The reason this is important is that this represents autolysosomes. In the rebuttal figure 'a', it is not necessary to use the same GFP/RFP signal as seen in HSC to define the gates, but they can be drawn using the same geometry (pattern) as the HSCs. The authors need to reanalyse their data with the same gating geometry as the Chua et al. 2023 paper, and as they have done in S1a.

Re: Thanks a lot for your points. According to your suggestion, we have modified the gating strategy according to CHUA et al. 2023 paper. The percentage of these 3 fractions appeared similar trend compared to our previous analysis (the following figure). The related data was updated in Figure 1b-1e and 1f as well as supplementary Figure 1b, 1g and 1h in the revised version.

Furthermore, we also reanalyzed the raw data related to Figure 4. All the details were modified in the revised manuscript. The alteration trend is similar to our previous analysis (the following figure). The data was updated in Figure 4a-4b, 4d-4e and supplementary Figure 4i, 4k, showing the same trend as our previous analysis.

All the data were updated in the revised manuscript. Because we have shown all the details were modified related Figure 1 and Figure 4 as well as supplementary Figure 1 and 4. Please See all the details in the manuscript.

My comment about line 128 is that GFP-RFP- are not obviously autolysosomes; if the authors measure the RFP/GFP MFI they will be low (rebuttal fig b) as they have been gated as low! The definition of autolysosomes consistent with the literature is RFP+GFP-, which a population gated in S1a (and in the Chua et al paper). To resolve this, the authors need to also stain for Lamp1 (as they have done in 1g in a bulk population) in the cell populations shown in rebuttal fig 'c'.

Re: Thanks a lot for your points. RFP⁺GFP⁻ fractions are similar to our previous fraction of RFP^{low}GFP^{low} by the new gating strategy according to your suggestion above. The purity of sorted cells was confirmed by immunostaining of GFP and RFP(following Figure a), similar to the normal sorting purity by flow cytometry.

The immunostaining for Lamp1 was performed in the cells from RFP⁺GFP⁺, RFP⁺GFP⁻, RFP⁻GFP⁻ populations of EC and/or pre-HSCs. In the following Figure b and c, Lamp1 was co-expressed highly in the RFP⁺ signals, in line with Figure 1g and supplementary Figure 1j, representing autolysosomes. Lamp1 signals are relative to the strength of RFP, higher in the RFP⁺GFP⁻ cells compared to that in the RFP⁺GFP⁺ cells of pre-HSC II. Based on our data and other report(Chua et al paper), RFP⁺GFP⁻ fraction represents autolysosomes. These figures were added to Figure 1f and supplementary Figure 1g-h and all the details were found in the revised manuscript.

Unfortunately there are also issues around the use of the bafilomycin experiments. Firstly, the duration of culture with bafilomycin is too long and it's not surprising there was toxicity. Bafilomycin should be used for a short time (2-4h). Secondly, reporting changes in the GFP/RFP ratio is uninformative as bafilomycin will raise lysosomal pH and therefore increase GFP signal notwithstanding any effect on autophagy.

Re: Thanks a lot for your points. As we mentioned in the last response, we tested the effects of Bafilomycin A1(Baf1) in explant cultures, in which AGM explants lay on the membranes supported by mash for air-liquid exchange. It is more difficult for Baf1 to enter into all cells of AGM tissues compared to that in single-cell layers. That is the reason that we have extended the culture time. Furthermore, the number of viable cells was calculated and there was no obvious alteration after Baf1 treatment as well as 3-MA, CQ treatment, in line with the percentage of viable cells after 6 hour-Baf1 treatment. Meanwhile, no alteration was found in the percentage of HEC and pre-HSC II after 6 hour-Baf1 treatment. Taken together, these data indicate that no toxicity existed in the 6-hour explant cultures at the existence of Baf1.

For the second point, you are right Baf1 would raise the lysosomal pH value and increase the GFP signals. Although our data confirmed this phenomenon, according to your comments, we have removed these data(supplementary Figure 4j in previous version) in the revised version.

The authors have also failed to understand the issue with using Atg5-LC3-GFP-RFP mice, and copy pasting an irrelevant answer about pH from an earlier section in the response to my comments about figure 4 is not satisfactory. Atg5 deletion does NOT block autophagosome lysosome fusion, it prevents autophagosome elongation by LC3 lipidation. They also do not understand my point about MTG and TMRE clashing with GFP/RFP and do not clarify that the mice lack LC3-GFP-RFP.

Re: 1) Thanks a lot for your critical point. I am so sorry to have misunderstood your points. Yes, you are correct that our description was not accurate. We have modified all the text related to Atg5-LC3-GFP-RFP cells. We have reanalyzed all the data by the new gating strategies according to your previous suggestion and modified all the details related to Figure 4. Please see all the details in the revised manuscript.

2) I am so sorry that I didn't explain the second point clearly in the last version of the manuscript. Because we detected the fluorescence signals of MTG(GFP channel) and TMRE(RFP channel) in the control and KO(Vec-Cre;Atg5^{fl/fl}) mouse models, which are not crossed with LC3-GFP-RFP, there is no clash existence between MTG and TMRE with GFP/RFP. Hopefully, we have explained it clearly.

In relation to my comments on pseudotime analysis, again there is no visibly discernible difference between WT and KO, and if the authors wish to claim there is, there needs to be some kind of statistical test done.

Re: Thanks a lot for your critical point. Indeed, the shift is not obvious in the EC and pre-HSCs between WT and KO, we have tried to display the data by area plot in the following figure, which looks more obvious in the pre-HSC I. The developmental process of HEC to pre-HSC I appears to be blocked compared to the control group, in line with our flow analysis data. We have moved these previous data into supplementary Figure 6c and added the area plot in Figure 6d. More details were modified in the revised manuscript.

Reviewer #2 (Remarks to the Author):

The authors have addressed the raised concerns.

Reviewer #1 (Remarks to the Author):

The authors have addressed my major concerns.